# On the Global Convergence of Training Deep Linear ResNets

**Difan Zou**
Department of Computer Science
University of California, Los Angeles
knowzou@cs.ucla.edu

**Philip M. Long**
Google
plong@google.com

**Quanquan Gu**
Department of Computer Science
University of California, Los Angeles
qgu@cs.ucla.edu

## ABSTRACT

We study the convergence of gradient descent (GD) and stochastic gradient descent (SGD) for training $L$-hidden-layer linear residual networks (ResNets). We prove that for training deep residual networks with certain linear transformations at input and output layers, which are fixed throughout training, both GD and SGD with zero initialization on all hidden weights can converge to the global minimum of the training loss. Moreover, when specializing to appropriate Gaussian random linear transformations, GD and SGD provably optimize wide enough deep linear ResNets. Compared with the global convergence result of GD for training standard deep linear networks (Du & Hu, 2019), our condition on the neural network width is sharper by a factor of $O(\kappa L)$, where $\kappa$ denotes the condition number of the covariance matrix of the training data. We further propose a modified identity input and output transformations, and show that a $(d + k)$-wide neural network is sufficient to guarantee the global convergence of GD/SGD, where $d, k$ are the input and output dimensions respectively.

## 1 INTRODUCTION

Despite the remarkable power of deep neural networks (DNNs) trained using stochastic gradient descent (SGD) in many machine learning applications, theoretical understanding of the properties of this algorithm, or even plain gradient descent (GD), remains limited. Many key properties of the learning process for such systems are also present in the idealized case of deep linear networks. For example, (a) the objective function is not convex; (b) errors back-propagate; and (c) there is potential for exploding and vanishing gradients. In addition to enabling study of systems with these properties in a relatively simple setting, analysis of deep linear networks also facilitates the scientific understanding of deep learning because using linear networks can control for the effect of architecture choices on the expressiveness of networks (Arora et al., 2018; Du & Hu, 2019). For these reasons, deep linear networks have received extensive attention in recent years.

One important line of theoretical investigation of deep linear networks concerns optimization landscape analysis (Kawaguchi, 2016; Hardt & Ma, 2016; Freeman & Bruna, 2016; Lu & Kawaguchi, 2017; Yun et al., 2018; Zhou & Liang, 2018), where major findings include that any critical point of a deep linear network with square loss function is either a global minimum or a saddle point, and identifying conditions on the weight matrices that exclude saddle points. Beyond landscape analysis, another research direction aims to establish convergence guarantees for optimization algorithms (e.g. GD, SGD) for training deep linear networks. Arora et al. (2018) studied the trajectory of gradient flow and showed that depth can help accelerate the optimization of deep linear networks. Ji & Telgarsky (2019); Gunasekar et al. (2018) investigated the implicit bias of GD for training deep linear networks and deep linear convolutional networks respectively. More recently, Bartlett et al. (2019); Arora et al. (2019a); Shamir (2018); Du & Hu (2019) analyzed the optimization trajectory of

GD for training deep linear networks and proved global convergence rates under certain assumptions on the training data, initialization, and neural network structure.

Inspired by the great empirical success of residual networks (ResNets), Hardt & Ma (2016) considered identity parameterizations in deep linear networks, i.e., parameterizing each layer's weight matrix as $\mathbf{I} + \mathbf{W}$, which leads to the so-called deep linear ResNets. In particular, Hardt & Ma (2016) established the existence of small norm solutions for deep residual networks with sufficiently large depth $L$, and proved that there are no critical points other than the global minimum when the maximum spectral norm among all weight matrices is smaller than $O(1/L)$. Motivated by this intriguing finding, Bartlett et al. (2019) studied the convergence rate of GD for training deep linear networks with identity initialization, which is equivalent to zero initialization in deep linear ResNets. They assumed whitened data and showed that GD can converge to the global minimum if (i) the training loss at the initialization is very close to optimal or (ii) the regression matrix $\mathbf{\Phi}$ is symmetric and positive definite. (In fact, they proved that, when $\mathbf{\Phi}$ is symmetric and has negative eigenvalues, GD for linear ResNets with zero-initialization does *not* converge.) Arora et al. (2019a) showed that GD converges under substantially weaker conditions, which can be satisfied by random initialization schemes. The convergence theory of *stochastic* gradient descent for training deep linear ResNets is largely missing; it remains unclear under which conditions SGD can be guaranteed to find the global minimum.

In this paper, we establish the global convergence of both GD and SGD for training deep linear ResNets without any condition on the training data. More specifically, we consider the training of $L$-hidden-layer deep linear ResNets with fixed linear transformations at input and output layers. We prove that under certain conditions on the input and output linear transformations, GD and SGD can converge to the global minimum of the training loss function. Moreover, when specializing to appropriate Gaussian random linear transformations, we show that, as long as the neural network is wide enough, both GD and SGD with zero initialization on all hidden weights can find the global minimum. There are two main ingredients of our proof: (i) establishing restricted gradient bounds and a smoothness property; and (ii) proving that these properties hold along the optimization trajectory and further lead to global convergence. We point out the second aspect is challenging especially for SGD due to the uncertainty of its optimization trajectory caused by stochastic gradients. We summarize our main contributions as follows:

- We prove the global convergence of GD and SGD for training deep linear ResNets. Specifically, we derive a generic condition on the input and output linear transformations, under which both GD and SGD with zero initialization on all hidden weights can find global minima. Based on this condition, one can design a variety of input and output transformations for training deep linear ResNets.

- When applying appropriate Gaussian random linear transformations, we show that as long as the neural network width satisfies $m = \Omega(kr\kappa^2)$, with high probability, GD can converge to the global minimum up to an $\epsilon$-error within $O(\kappa \log(1/\epsilon))$ iterations, where $k, r$ are the output dimension and the rank of training data matrix $\mathbf{X}$ respectively, and $\kappa = \|\mathbf{X}\|_2^2 / \sigma_r^2(\mathbf{X})$ denotes the condition number of the covariance matrix of the training data. Compared with previous convergence results for training deep linear networks from Du & Hu (2019), our condition on the neural network width is independent of the neural network depth $L$, and is strictly better by a factor of $O(L\kappa)$.

- Using the same Gaussian random linear transformations, we also establish the convergence guarantee of SGD for training deep linear ResNets. We show that if the neural network width satisfies $m = \widetilde{\Omega}\big(kr\kappa^2 \log^2(1/\epsilon) \cdot n^2/B^2\big)$, with constant probability, SGD can converge to the global minimum up to an $\epsilon$-error within $\widetilde{O}\big(\kappa^2 \epsilon^{-1} \log(1/\epsilon) \cdot n/B\big)$ iterations, where $n$ is the training sample size and $B$ is the minibatch size of stochastic gradient. This is the first global convergence rate of SGD for training deep linear networks. Moreover, when the global minimum of the training loss is 0, we prove that SGD can further achieve linear rate of global convergence, and the condition on the neural network width does not depend on the target error $\epsilon$.

As alluded to above, we analyze networks with $d$ inputs, $k$ outputs, and $m \geqslant \max\{d, k\}$ nodes in each hidden layer. Linear transformations that are fixed throughout training map the inputs to the first hidden layer, and the last hidden layer to the outputs. We prove that our bounds hold with high probability when these input and output transformations are randomly generated by Gaussian distributions. If, instead, the input transformation simply copies the inputs onto the first $d$ compo-

nents of the first hidden layer, and the output transformation takes the first $k$ components of the last hidden layer, then our analysis does not provide a guarantee. There is a good reason for this: a slight modification of a lower bound argument from Bartlett et al. (2019) demonstrates that GD may fail to converge in this case. However, we describe a similarly simple, deterministic, choice of input and output transformations such that wide enough networks *always* converge. The resulting condition on the network width is weaker than that for Gaussian random transformations, and thus improves on the corresponding convergence guarantee for linear networks, which, in addition to requiring wider networks, only hold with high probability for random transformations.

## 1.1 ADDITIONAL RELATED WORK

In addition to what we discussed above, a large bunch of work focusing on the optimization of neural networks with nonlinear activation functions has emerged. We will briefly review them in this subsection.

It is widely believed that the training loss landscape of nonlinear neural networks is highly nonconvex and nonsmooth (e.g., neural networks with ReLU/LeakyReLU activation), thus it is fundamentally difficult to characterize the optimization trajectory and convergence performance of GD and SGD. Some early work (Andoni et al., 2014; Daniely, 2017) showed that wide enough (polynomial in sample size $n$) neural networks trained by GD/SGD can learn a class of continuous functions (e.g., polynomial functions) in polynomial time. However, those works only consider training some of the neural network weights rather than all of them (e.g., the input and output layers) [1]. In addition, a series of papers investigated the convergence of gradient descent for training shallow networks (typically 2-layer networks) under certain assumptions on the training data and initialization scheme (Tian, 2017; Du et al., 2018b; Brutzkus et al., 2018; Zhong et al., 2017; Li & Yuan, 2017; Zhang et al., 2018). However, the assumptions made in these works are rather strong and not consistent with practice. For example, Tian (2017); Du et al. (2018b); Zhong et al. (2017); Li & Yuan (2017); Zhang et al. (2018) assumed that the label of each training data is generated by a teacher network, which has the same architecture as the learned network. Brutzkus et al. (2018) assumed that the training data is linearly separable. Li & Liang (2018) addressed this drawback; they proved that for two-layer ReLU network with cross-entropy loss, as long as the neural network is sufficiently wide, under mild assumptions on the training data SGD with commonly-used Gaussian random initialization can achieve nearly zero expected error. Du et al. (2018c) proved the similar results of GD for training two-layer ReLU networks with square loss. Beyond shallow neural networks, Allen-Zhu et al. (2019); Du et al. (2019); Zou et al. (2019) generalized the global convergence results to multi-layer over-parameterized ReLU networks. Chizat et al. (2019) showed that training over-parameterized neural networks actually belongs to a so-called "lazy training" regime, in which the model behaves like its linearization around the initialization. Furthermore, the parameter scaling is more essential than over-paramterization to make the model learning within the "lazy training" regime. Along this line of research, several follow up works have been conducted. Oymak & Soltanolkotabi (2019); Zou & Gu (2019); Su & Yang (2019); Kawaguchi & Huang (2019) improved the convergence rate and over-parameterization condition for both shallow and deep networks. Arora et al. (2019b) showed that training a sufficiently wide deep neural network is almost equivalent to kernel regression using neural tangent kernel (NTK), proposed in Jacot et al. (2018). Allen-Zhu et al. (2019); Du et al. (2019); Zhang et al. (2019) proved the global convergence for training deep ReLU ResNets. Frei et al. (2019) proved the convergence of GD for training deep ReLU ResNets under an over-parameterization condition that is only logarithmic in the depth of the network, which partially explains why deep residual networks are preferable to fully connected ones. However, all the results in Allen-Zhu et al. (2019); Du et al. (2019); Zhang et al. (2019); Frei et al. (2019) require a very stringent condition on the network width, which typically has a high-degree polynomial dependence on the training sample size $n$. Besides, the results in Allen-Zhu et al. (2019); Zhang et al. (2019) also require that all data points are separated by a positive distance and have unit norm. As shown in Du & Hu (2019) and will be proved in this paper, for deep linear (residual) networks, there is no assumption on the training data, and the condition on the network width is significantly milder, which is independent of the sample size $n$. While achieving a stronger result for linear networks than for nonlinear ones is not surprising, we believe that our analysis, conducted in the idealized deep linear case, can provide useful insights to understand optimization in the nonlinear case.

---

[1] In Daniely (2017), the weight changes in all hidden layers make negligible contribution to the final output, thus can be approximately treated as only training the output layer.

Two concurrent works analyze gradient descent applied to deep linear (residual) networks (Hu et al., 2020; Wu et al., 2019). Hu et al. (2020) consider deep linear networks with orthogonal initialization, and Wu et al. (2019) consider zero initialization on the last layer and identity initialization for the rest of the layers, which are similar to our setting. However, there are several differences between their work and ours. One major difference is that Hu et al. (2020) and Wu et al. (2019) only prove global convergence for GD, but our results cover both GD and SGD. In addition, Hu et al. (2020) focuses on proving the global convergence of GD for sufficiently wide networks, while we provide a generic condition on the input and output linear transformations for ensuring global convergence. Wu et al. (2019) assumes whitened data and proves a $O(L^3 \log(1/\epsilon))$ bound on the number of iterations required for GD to converge, where we establish a $O(\log(1/\epsilon))^2$ bound.[2]

## 1.2 NOTATION.

We use lower case, lower case bold face, and upper case bold face letters to denote scalars, vectors and matrices respectively. For a positive integer, we denote the set $\{1, \ldots, k\}$ by $[k]$. Given a vector $\mathbf{x}$, we use $\|\mathbf{x}\|_2$ to denote its $\ell_2$ norm. We use $N(\mu, \sigma^2)$ to denote the Gaussian distribution with mean $\mu$ and variance $\sigma^2$. Given a matrix $\mathbf{X}$, we denote $\|\mathbf{X}\|_F$, $\|\mathbf{X}\|_2$ and $\|\mathbf{X}\|_{2,\infty}$ as its Frobenious norm, spectral norm and $\ell_{2,\infty}$ norm (maximum $\ell_2$ norm over its columns), respectively. In addition, we denote by $\sigma_{\min}(\mathbf{X})$, $\sigma_{\max}(\mathbf{X})$ and $\sigma_r(\mathbf{X})$ the smallest, largest and $r$-th largest singular values of $\mathbf{X}$ respectively. For a square matrix $\mathbf{A}$, we denote by $\lambda_{\min}(\mathbf{A})$ and $\lambda_{\max}(\mathbf{A})$ the smallest and largest eigenvalues of $\mathbf{A}$ respectively. For two sequences $\{a_k\}_{k \geq 0}$ and $\{b_k\}_{k \geq 0}$, we say $a_k = O(b_k)$ if $a_k \leq C_1 b_k$ for some absolute constant $C_1$, and use $a_k = \Omega(b_k)$ if $a_k \geq C_2 b_k$ for some absolute constant $C_2$. Except the target error $\epsilon$, we use $\widetilde{O}(\cdot)$ and $\widetilde{\Omega}(\cdot)$ to hide the logarithmic factors in $O(\cdot)$ and $\Omega(\cdot)$ respectively.

## 2 PROBLEM SETUP

**Model.** In this work, we consider deep linear ResNets defined as follows:

$$f_{\mathbf{W}}(\mathbf{x}) = \mathbf{B}(\mathbf{I} + \mathbf{W}_L) \ldots (\mathbf{I} + \mathbf{W}_1)\mathbf{A}\mathbf{x},$$

where $\mathbf{x} \in \mathbb{R}^d$ is the input, $f_{\mathbf{W}}(\mathbf{x}) \in \mathbb{R}^k$ is the corresponding output, $\mathbf{A} \in \mathbb{R}^{m \times d}, \mathbf{B} \in \mathbb{R}^{k \times m}$ denote the weight matrices of input and output layers respectively, and $\mathbf{W}_1, \ldots, \mathbf{W}_L \in \mathbb{R}^{m \times m}$ denote the weight matrices of all hidden layers. The formulation of ResNets in our paper is different from that in Hardt & Ma (2016); Bartlett et al. (2019), where the hidden layers have the same width as the input and output layers. In our formulation, we allow the hidden layers to be wider by choosing the dimensions of $\mathbf{A}$ and $\mathbf{B}$ appropriately.

**Loss Function.** Let $\{(\mathbf{x}_i, \mathbf{y}_i)\}_{i=1,\ldots,n}$ be the training dataset, $\mathbf{X} = (\mathbf{x}_1, \ldots, \mathbf{x}_n) \in \mathbb{R}^{d \times n}$ be the input data matrix and $\mathbf{Y} = (\mathbf{y}_1, \ldots, \mathbf{y}_n) \in \mathbb{R}^{k \times n}$ be the corresponding output label matrix. We assume the data matrix $\mathbf{X}$ is of rank $r$, where $r$ can be smaller than $d$. Let $\mathbf{W} = \{\mathbf{W}_1, \ldots, \mathbf{W}_L\}$ be the collection of weight matrices of all hidden layers. For an example $(\mathbf{x}, \mathbf{y})$, we consider the square loss defined by

$$\ell(\mathbf{W}; \mathbf{x}, \mathbf{y}) = \frac{1}{2}\|f_{\mathbf{W}}(\mathbf{x}) - \mathbf{y}\|_2^2.$$

Then the training loss over the training dataset takes the following form

$$L(\mathbf{W}) := \sum_{i=1}^{n} \ell(\mathbf{W}; \mathbf{x}_i, \mathbf{y}_i) = \frac{1}{2}\|\mathbf{B}(\mathbf{I} + \mathbf{W}_L) \cdots (\mathbf{I} + \mathbf{W}_1)\mathbf{A}\mathbf{X} - \mathbf{Y}\|_F^2.$$

**Algorithm.** Similar to Allen-Zhu et al. (2019); Zhang et al. (2019), we consider algorithms that only train the weights $\mathbf{W}$ for hidden layers while leaving the input and output weights $\mathbf{A}$ and $\mathbf{B}$ unchanged throughout training. For hidden weights, we follow the similar idea in Bartlett et al. (2019) and adopt zero initialization (which is equivalent to identity initialization for standard linear network). We would also like to point out that at the initialization, all the hidden layers automatically satisfy the so-called balancedness condition (Arora et al., 2018; 2019a; Du et al., 2018a). The optimization algorithms, including GD and SGD, are summarized in Algorithm 1.

---

[2]Considering whitened data immediately gives $\kappa = 1$.

---

**Algorithm 1** (Stochastic) Gradient descent with zero initialization

---

1: **input:** Training data $\{\mathbf{x}_i, \mathbf{y}_i\}_{i \in [n]}$, step size $\eta$, total number of iterations $T$, minibatch size $B$, input and output weight matrices $\mathbf{A}$ and $\mathbf{B}$.

2: **initialization:** For all $l \in [L]$, each entry of weight matrix $\mathbf{W}_l^{(0)}$ is initialized as $\mathbf{0}$.

————————————————————— **Gradient Descent** —————————————————————

3: **for** $t = 0, \ldots, T-1$ **do**

4: $\quad \mathbf{W}_l^{(t+1)} = \mathbf{W}_l^{(t)} - \eta \nabla_{\mathbf{W}_l} L(\mathbf{W}^{(t)})$ for all $l \in [L]$

5: **end for**

6: **output:** $\mathbf{W}^{(T)}$

————————————————— **Stochastic Gradient Descent** —————————————————

7: **for** $t = 0, \ldots, T-1$ **do**

8: $\quad$ Uniformly sample a subset $\mathcal{B}^{(t)}$ of size $B$ from training data without replacement.

9: $\quad$ For all $\ell \in [L]$, compute the stochastic gradient $\mathbf{G}_l^{(t)} = \frac{n}{B} \sum_{i \in \mathcal{B}^{(t)}} \nabla_{\mathbf{W}_l} \ell(\mathbf{W}^{(t)}; \mathbf{x}_i, \mathbf{y}_i)$

10: $\quad$ For all $l \in [L]$, $\mathbf{W}_l^{(t+1)} = \mathbf{W}_l^{(t)} - \eta \mathbf{G}_l^{(t)}$

11: **end for**

12: **output:** $\{\mathbf{W}^{(t)}\}_{t=0,\ldots,T}$

---

## 3 MAIN THEORY

It is clear that the expressive power of deep linear ResNets is identical to that of simple linear model, which implies that the global minima of deep linear ResNets cannot be smaller than that of linear model. Therefore, our focus is to show that GD/SGD can converge to a point $\mathbf{W}^*$ with

$$L(\mathbf{W}^*) = \min_{\mathbf{\Theta} \in \mathbb{R}^{k \times d}} \frac{1}{2} \|\mathbf{\Theta X} - \mathbf{Y}\|_F^2,$$

which is exactly the global minimum of the linear regression problem. It what follows, we will show that with appropriate input and output transformations, both GD and SGD can converge to the global minimum.

### 3.1 CONVERGENCE GUARANTEE OF GRADIENT DESCENT

The following theorem establishes the global convergence of GD for training deep linear ResNets.

**Theorem 3.1.** There are absolute constants $C$ and $C_1$ such that, if the input and output weight matrices satisfy

$$\frac{\sigma_{\min}^2(\mathbf{A})\sigma_{\min}^2(\mathbf{B})}{\|\mathbf{A}\|_2 \|\mathbf{B}\|_2} \geqslant C \frac{\|\mathbf{X}\|_2 \big(L(\mathbf{W}^{(0)}) - L(\mathbf{W}^*)\big)^{1/2}}{\sigma_r^2(\mathbf{X})}$$

and the step size satisfies

$$\eta \leqslant C_1 \cdot \frac{1}{L\|\mathbf{A}\|_2 \|\mathbf{B}\|_2 \|\mathbf{X}\|_2 \cdot \big(\sqrt{L(\mathbf{W}^{(0)})} + \|\mathbf{A}\|_2 \|\mathbf{B}\|_2 \|\mathbf{X}\|_2\big)},$$

then for all iterates of GD in Algorithm 1, it holds that

$$L(\mathbf{W}^{(t)}) - L(\mathbf{W}^*) \leqslant \left(1 - \frac{\eta L \sigma_{\min}^2(\mathbf{A})\sigma_{\min}^2(\mathbf{B})\sigma_r^2(\mathbf{X})}{e}\right)^t \cdot \big(L(\mathbf{W}^{(0)}) - L(\mathbf{W}^*)\big).$$

**Remark 3.2.** Theorem 3.1 can imply the convergence result in Bartlett et al. (2019). Specifically, in order to turn into the setting considered in Bartlett et al. (2019), we choose $m = d = k$, $\mathbf{A} = \mathbf{I}$, $\mathbf{B} = \mathbf{I}$, $L(\mathbf{W}^*) = 0$ and $\mathbf{X}\mathbf{X}^\top = \mathbf{I}$. Then it can be easily observed that the condition in Theorem 3.1 becomes $L(\mathbf{W}^{(0)}) - L(\mathbf{W}^*) \leqslant C^{-2}$. This implies that the global convergence can be established as long as $L(\mathbf{W}^{(0)}) - L(\mathbf{W}^*)$ is smaller than some constant, which is equivalent to the condition proved in Bartlett et al. (2019).

In general, $L(\mathbf{W}^{(0)}) - L(\mathbf{W}^*)$ can be large and thus the setting considered in Bartlett et al. (2019) may not be able to guarantee global convergence. Therefore, it is natural to ask in which setting

the condition on $\mathbf{A}$ and $\mathbf{B}$ in Theorem 3.1 can be satisfied. Here we provide one possible choice which is commonly used in practice (another viable choices can be found in Section 4). We use Gaussian random input and output transformations, i.e., each entry in $\mathbf{A}$ is independently generated from $N(0, 1/m)$ and each entry in $\mathbf{B}$ is generated from $N(0, 1/k)$. Based on this choice of transformations, we have the following proposition that characterizes the quantity of the largest and smallest singular values of $\mathbf{A}$ and $\mathbf{B}$, and the training loss at the initialization (i.e., $L(\mathbf{W}^{(0)})$). The following proposition is proved in Section A.2.

**Proposition 3.3.** In Algorithm 1, if each entry in $\mathbf{A}$ is independently generated from $N(0, \alpha^2)$ and each entry in $\mathbf{B}$ is independently generated from $N(0, \beta^2)$, then if $m \geqslant C \cdot (d + k + \log(1/\delta))$ for some absolute constant $C$, with probability at least $1 - \delta$, it holds that

$$\sigma_{\min}(\mathbf{A}) = \Omega(\alpha\sqrt{m}), \ \sigma_{\max}(\mathbf{A}) = O(\alpha\sqrt{m}), \quad \sigma_{\min}(\mathbf{B}) = \Omega(\beta\sqrt{m}), \ \sigma_{\max}(\mathbf{B}) = O(\beta\sqrt{m}),$$

$$\text{and} \quad L(\mathbf{W}^{(0)}) \leqslant O(\alpha^2\beta^2 km \log(n/\delta)\|\mathbf{X}\|_F^2 + \|\mathbf{Y}\|_F^2).$$

Then based on Theorem 3.1 and Proposition 3.3, we provide the following corollary, proved in Section 3.4, which shows that GD is able to achieve global convergence if the neural network is wide enough.

**Corollary 3.4.** Suppose $\|\mathbf{Y}\|_F = O(\|\mathbf{X}\|_F)$. Then using Gaussian random input and output transformations in Proposition 3.3 with $\alpha = \beta = 1$, if the neural network width satisfies $m = \Omega(\max\{kr\kappa^2 \log(n/\delta), k + d + \log(1/\delta)\})$ then, with probability at least $1 - \delta$, the output of GD in Algorithm 1 achieves training loss at most $L(\mathbf{W}^*) + \epsilon$ within $T = O(\kappa \log(1/\epsilon))$ iterations, where $\kappa = \|\mathbf{X}\|_2^2/\sigma_r^2(\mathbf{X})$ denotes the condition number of the covariance matrix of training data.

**Remark 3.5.** For standard deep linear networks, Du & Hu (2019) proved that GD with Gaussian random initialization can converge to a $\epsilon$-suboptimal global minima within $T = \Omega(\kappa \log(1/\epsilon))$ iterations if the neural network width satisfies $m = O(Lkr\kappa^3 + d)$. In stark contrast, training deep linear ResNets achieves the same convergence rate as training deep linear networks and linear regression, while the condition on the neural network width is strictly milder than that for training standard deep linear networks by a factor of $O(L\kappa)$. This improvement may in part validate the empirical advantage of deep ResNets.

## 3.2 Convergence Guarantee of Stochastic Gradient Descent

The following theorem establishes the global convergence of SGD for training deep linear ResNets.

**Theorem 3.6.** There are absolute constants $C$, $C_1$ and $C_2$, such for any $0 < \delta \leqslant 1/6$ and $\epsilon > 0$, if the input and output weight matrices satisfy

$$\frac{\sigma_{\min}^2(\mathbf{A})\sigma_{\min}^2(\mathbf{B})}{\|\mathbf{A}\|_2\|\mathbf{B}\|_2} \geqslant C \cdot \frac{n\|\mathbf{X}\|_2 \cdot \log(L(\mathbf{W}^{(0)})/\epsilon)}{B\sigma_r^2(\mathbf{X})} \cdot \sqrt{L(\mathbf{W}^{(0)})},$$

and the step size and maximum iteration number are set as

$$\eta \leqslant C_1 \cdot \frac{B\sigma_{\min}^2(\mathbf{A})\sigma_{\min}^2(\mathbf{B})\sigma_r^2(\mathbf{X})}{Ln\|\mathbf{A}\|_2^4\|\mathbf{B}\|_2^4\|\mathbf{X}\|_2^2} \cdot \min\left\{\frac{\epsilon}{\|\mathbf{X}\|_{2,\infty}^2 L(\mathbf{W}^*)}, \frac{B}{n\|\mathbf{X}\|_2^2 \cdot \log(T/\delta)\log(L(\mathbf{W}^{(0)})/\epsilon)}\right\},$$

$$T = C_2 \cdot \frac{1}{\eta L\sigma_{\min}^2(\mathbf{A})\sigma_{\min}^2(\mathbf{B})\sigma_r^2(\mathbf{X})} \cdot \log\left(\frac{L(\mathbf{W}^{(0)}) - L(\mathbf{W}^*)}{\epsilon}\right),$$

then with probability[3] at least $1/2$ (with respect to the random choices of mini batches), SGD in Algorithm 1 can find a network that achieves training loss at most $L(\mathbf{W}^*) + \epsilon$.

By combining Theorem 3.6 and Proposition 3.3, we can show that as long as the neural network is wide enough, SGD can achieve global convergence. Specifically, we provide the condition on the neural network width and the iteration complexity of SGD in the following corollary.

**Corollary 3.7.** Suppose $\|\mathbf{Y}\|_F = O(\|\mathbf{X}\|_F)$. Then using Gaussian random input and output transformations in Proposition 3.3 with $\alpha = \beta = 1$, for sufficiently small $\epsilon > 0$, if the neural network width satisfies $m = \widetilde{\Omega}(kr\kappa^2 \log^2(1/\epsilon) \cdot n^2/B^2 + d)$, with constant probability, SGD in Algorithm 1 can find a point that achieves training loss at most $L(\mathbf{W}^*) + \epsilon$ within $T = \widetilde{O}(\kappa^2\epsilon^{-1}\log(1/\epsilon) \cdot n/B)$ iterations.

---

[3]One can boost this probability to $1 - \delta$ by independently running $\log(1/\delta)$ copies of SGD in Algorithm 1.

From Corollaries 3.7 and 3.4, we can see that compared with the convergence guarantee of GD, the condition on the neural network width for SGD is worse by a factor of $\widetilde{O}(n^2 \log^2(1/\epsilon)/B^2)$ and the iteration complexity is higher by a factor of $\widetilde{O}(\kappa \epsilon^{-1} \cdot n/B)$. This is because for SGD, its trajectory length contains high uncertainty, and thus we need stronger conditions on the neural network in order to fully control it.

We further consider the special case that $L(\mathbf{W}^*) = 0$, which implies that there exists a ground truth matrix $\mathbf{\Phi}$ such that for each training data point $(\mathbf{x}_i, \mathbf{y}_i)$ we have $\mathbf{y}_i = \mathbf{\Phi} \mathbf{x}_i$. In this case, we have the following theorem, which shows that SGD can attain a linear rate to converge to the global minimum.

**Theorem 3.8.** There are absolute constants $C$, and $C_1$ such that for any $0 < \delta < 1$, if the input and output weight matrices satisfy

$$\frac{\sigma^2_{\min}(\mathbf{A})\sigma^2_{\min}(\mathbf{B})}{\|\mathbf{A}\|_2\|\mathbf{B}\|_2} \geqslant C \cdot \frac{n\|\mathbf{X}\|_2}{B\sigma^2_r(\mathbf{X})} \cdot \sqrt{L(\mathbf{W}^{(0)})},$$

and the step size is set as

$$\eta \leqslant C_1 \cdot \frac{B^2 \sigma^2_{\min}(\mathbf{A})\sigma^2_{\min}(\mathbf{B})\sigma^2_r(\mathbf{X})}{Ln^2\|\mathbf{A}\|^4_2\|\mathbf{B}\|^4_2\|\mathbf{X}\|^4_2 \cdot \log(T/\delta)},$$

for some maximum iteration number $T$, then with probability at least $1 - \delta$, the following holds for all $t \leqslant T$,

$$L(\mathbf{W}^{(t)}) \leqslant 2L(\mathbf{W}^{(0)}) \cdot \left(1 - \frac{\eta L \sigma^2_{\min}(\mathbf{A})\sigma^2_{\min}(\mathbf{B})\sigma^2_r(\mathbf{X})}{e}\right)^t.$$

Similarly, using Gaussian random transformations in Proposition 3.3, we show that SGD can achieve global convergence for wide enough deep linear ResNets in the following corollary.

**Corollary 3.9.** Suppose $\|\mathbf{Y}\|_F = O(\|\mathbf{X}\|_F)$. Then using Gaussian random transformations in Proposition 3.3 with $\alpha = \beta = 1$, for any $\epsilon \leqslant \widetilde{O}\big(B\|\mathbf{X}\|^2_{2,\infty}/(n\|\mathbf{X}\|^2_2)\big)$, if the neural network width satisfies $m = \widetilde{\Omega}\big(kr\kappa^2 \cdot n^2/B^2 + d\big)$, with high probability, SGD in Algorithm 1 can find a network that achieves training loss at most $\epsilon$ within $T = \widetilde{O}\big(\kappa^2 \log(1/\epsilon) \cdot n^2/B^2\big)$ iterations.

# 4 DISCUSSION ON DIFFERENT INPUT AND OUTPUT LINEAR TRANSFORMATIONS

In this section, we will discuss several different choices of linear transformations at input and output layers and their effects to the convergence performance. For simplicity, we will only consider the condition for GD.

As we stated in Subsection 3.1, GD converges if the input and output weight matrices $\mathbf{A}$ and $\mathbf{B}$

$$\frac{\sigma^2_{\min}(\mathbf{A})\sigma^2_{\min}(\mathbf{B})}{\|\mathbf{A}\|_2\|\mathbf{B}\|_2} \geqslant C \cdot \frac{\|\mathbf{X}\|_2}{\sigma^2_r(\mathbf{X})} \cdot \big(L(\mathbf{W}^{(0)}) - L(\mathbf{W}^*)\big)^{1/2}. \tag{4.1}$$

Then it is interesting to figure out what kind of choice of $\mathbf{A}$ and $\mathbf{B}$ can satisfy this condition. In Proposition 3.3, we showed that Gaussian random transformations (i.e., each entry of $\mathbf{A}$ and $\mathbf{B}$ is generated from certain Gaussian distribution) satisfy this condition with high probability, so that GD converges. Here we will discuss the following two other transformations.

**Identity transformations.** We first consider the transformations that $\mathbf{A} = \big[\mathbf{I}_{d \times d}, \mathbf{0}_{d \times (m-d)}\big]^\top$ and $\mathbf{B} = \sqrt{m/k} \cdot \big[\mathbf{I}_{k \times k}, \mathbf{0}_{k \times (m-k)}\big]$. which is equivalent to the setting in Bartlett et al. (2019) when $m = k = d$. Then it is clear that

$$\sigma_{\min}(\mathbf{B}) = \sigma_{\max}(\mathbf{B}) = \sqrt{m/k} \quad \text{and} \quad \sigma_{\min}(\mathbf{A}) = \sigma_{\max}(\mathbf{A}) = 1.$$

Now let us consider $L(\mathbf{W}^{(0)})$. By our choices of $\mathbf{B}$ and $\mathbf{A}$ and zero initialization on weight matrices in hidden layers, in the case that $d = k$, we have

$$L(\mathbf{W}^{(0)}) = \frac{1}{2}\|\mathbf{B}\mathbf{A}\mathbf{X} - \mathbf{Y}\|^2_F = \frac{1}{2}\big\|\sqrt{m/k}\mathbf{X} - \mathbf{Y}\big\|^2_F.$$

We remark that $\left\|\sqrt{m/k}\mathbf{X} - \mathbf{Y}\right\|_F^2/2$ could be as big as $\frac{1}{2}\left(m\|\mathbf{X}\|_F^2/k + \|\mathbf{Y}\|_F^2\right)$ (for example, when $\mathbf{X}$ and $\mathbf{Y}$ are orthogonal). Then plugging these results into (4.1), the condition on $\mathbf{A}$ and $\mathbf{B}$ becomes

$$\sqrt{m/k} \geqslant C \cdot \frac{\|\mathbf{X}\|_2}{\sigma_r^2(\mathbf{X})} \cdot \left(\frac{1}{2}\left(m\|\mathbf{X}\|_F^2/k + \|\mathbf{Y}\|_F^2\right) - L(\mathbf{W}^*)\right)^{1/2} \geqslant C \cdot \frac{\|\mathbf{X}\|_2}{\sigma_r^2(\mathbf{X})} \cdot \sqrt{\frac{m\|\mathbf{X}\|_F^2}{2k}},$$

where the second inequality is due to the fact that $L(\mathbf{W}^*) \leqslant \|\mathbf{Y}\|_F^2/2$. Then it is clear if $\|\mathbf{X}\|_F \geqslant \sqrt{2}/C$, the above inequality cannot be satisfied for any choice of $m$, since it will be cancelled out on both sides of the inequality. Therefore, in such cases, our bound does not guarantee that GD achieves global convergence. Thus, it is consistent with the non-convergence results in (Bartlett et al., 2019). Note that replacing the scaling factor $\sqrt{m/k}$ in the definition of $\mathbf{B}$ with any other function of $d$, $k$ and $m$ would not help.

**Modified identity transformations.** In fact, we show that a different type of identity transformations of $\mathbf{A}$ and $\mathbf{B}$ can satisfy the condition (4.1). Here we provide one such example. Assuming $m \geqslant d + k$, we can construct two sets $\mathcal{S}_1, \mathcal{S}_2 \subset [m]$ satisfying $|\mathcal{S}_1| = d$, $|\mathcal{S}_2| = k$ and $\mathcal{S}_1 \cap \mathcal{S}_2 = \varnothing$. Let $\mathcal{S}_1 = \{i_1, \ldots, i_d\}$ and $\mathcal{S}_2 = \{j_1, \ldots, j_k\}$. Then we construct matrices $\mathbf{A}$ and $\mathbf{B}$ as follows:

$$\mathbf{A}_{ij} = \left\{\begin{array}{ll} 1 & (i,j) = (i_j, j) \\ 0 & \text{otherwise} \end{array}\right. \quad \mathbf{B}_{ij} = \left\{\begin{array}{ll} \alpha & (i,j) = (i, j_i) \\ 0 & \text{otherwise} \end{array}\right.$$

where $\alpha$ is a parameter which will be specified later. In this way, it can be verified that $\mathbf{B}\mathbf{A} = \mathbf{0}$, $\sigma_{\min}(\mathbf{A}) = \sigma_{\max}(\mathbf{A}) = 1$, and $\sigma_{\min}(\mathbf{B}) = \sigma_{\max}(\mathbf{B}) = \alpha$. Thus it is clear that the initial training loss satisfies $L(\mathbf{W}^{(0)}) = \|\mathbf{Y}\|_F^2/2$. Then plugging these results into (4.1), the condition on $\mathbf{A}$ and $\mathbf{B}$ can be rewritten as

$$\alpha \geqslant C \cdot \frac{\|\mathbf{X}\|_2}{\sigma_r^2(\mathbf{X})} \cdot \left(\|\mathbf{Y}\|_F^2/2 - L(\mathbf{W}^*)\right)^{1/2}.$$

The R.H.S. of the above inequality does not depend on $\alpha$, which implies that we can choose sufficiently large $\alpha$ to make this inequality hold. Thus, GD can be guaranteed to achieve the global convergence. Moreover, it is worth noting that using modified identity transformation, a neural network with $m = d + k$ suffices to guarantee the global convergence of GD. We further remark that similar analysis can be extended to SGD.

## 5 EXPERIMENTS

In this section, we conduct various experiments to verify our theory on synthetic data, including i) comparison between different input and output transformations and ii) comparison between training deep linear ResNets and standard linear networks.

### 5.1 DIFFERENT INPUT AND OUTPUT TRANSFORMATIONS

To validate our theory, we performed simple experiment on 10-d synthetic data. Specifically, we randomly generate $\mathbf{X} \in \mathbb{R}^{10 \times 1000}$ from a standard normal distribution and set $\mathbf{Y} = -\mathbf{X} + 0.1 \cdot \mathbf{E}$, where each entry in $\mathbf{E}$ is independently generated from standard normal distribution. Consider 10-hidden-layer linear ResNets, we apply three input and output transformations including *identity transformations*, *modified identity transformations* and *random transformations*. We evaluate the convergence performances for these three choices of transformations and report the results in Figures 1(a)-1(b), where we consider two cases $m = 40$ and $m = 200$. It can be clearly observed that gradient descent with identity initialization gets stuck, but gradient descent with modified identity initialization or random initialization converges well. This verifies our theory. It can be also observed that modified identity initialization can lead to slightly faster convergence rate as its initial training loss can be smaller. In fact, with identity transformations in this setting, only the first 10 entries of the $m$ hidden variables in each layer ever take a non-zero value, so that, no matter how large $m$ is, effectively, $m = 10$, and the lower bound of Bartlett et al. (2019) applies.

### 5.2 COMPARISON WITH STANDARD DEEP LINEAR NETWORKS

Then we compare the convergence performances with that of training standard deep linear networks. Specifically, we adopt the same training data generated in Section 5.1 and consider training $L$-hidden-layer neural network with fixed width $m$. The convergence results are displayed in Figures

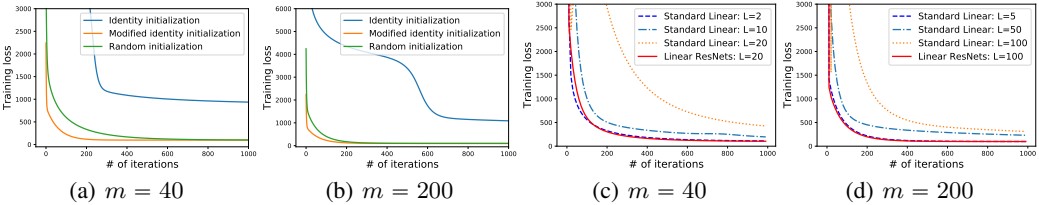

(a) $m = 40$      (b) $m = 200$      (c) $m = 40$      (d) $m = 200$

Figure 1: (a)-(b):Convergence performances for three input and output transformations on a 10-hidden-layer linear ResNets. (c)-(d) Comparison between the convergence performances of training deep linear ResNets with zero initialization on hidden weights and standard deep linear network with Gaussian random initialization on hidden weights, where the input and output weights are generated by random initialization, and remain fixed throughout the training.

1(c)-1(d), where we consider different choices of $L$. For training linear ResNets, we found that the convergence performances are quite similar for different $L$, thus we only plot the convergence result for the largest one (e.g., $L = 20$ for $m = 40$ and $L = 100$ for $m = 200$). However, it can be observed that for training standard linear networks, the convergence performance becomes worse as the depth increases. This is consistent with the theory as our condition on the neural network width is $m = O(kr\kappa^2)$ (please refer to Corollary 3.4), which has no dependency in $L$, while the condition for training standard linear network is $m = O(Lkr\kappa^3)$ (Du & Hu, 2019), which is linear in $L$.

## 6 CONCLUSION

In this paper, we proved the global convergence of GD and SGD for training deep linear ResNets with square loss. More specifically, we considered fixed linear transformations at both input and output layers, and proved that under certain conditions on the transformations, GD and SGD with zero initialization on all hidden weights can converge to the global minimum. In addition, we further proved that when specializing to appropriate Gaussian random linear transformations, GD and SGD can converge as long as the neural network is wide enough. Compared with the convergence results of GD for training standard deep linear networks, our condition on the neural network width is strictly milder. Our analysis can be generalized to prove similar results for different loss functions such as cross-entropy loss, and can potentially provide meaningful insights to the convergence analysis of deep non-linear ResNets.

ACKNOWLEDGEMENT

We thank the anonymous reviewers and area chair for their helpful comments. This work was initiated when Q. Gu and P. Long attended the summer program on the Foundations of Deep Learning at the Simons Institute for the Theory of Computing. D. Zou and Q. Gu were sponsored in part by the National Science Foundation CAREER Award IIS-1906169, BIGDATA IIS-1855099, and Salesforce Deep Learning Research Award. The views and conclusions contained in this paper are those of the authors and should not be interpreted as representing any funding agencies.

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

## A  PROOF OF MAIN THEOREMS

We first provide the following lemma which proves upper and lower bounds on $\|\nabla_{\mathbf{W}_l} L(\mathbf{W})\|_F^2$ when $\mathbf{W}$ is staying inside a certain region. Its proof is in Section B.1.

**Lemma A.1.** For any weight matrices satisfying $\max_{l \in [L]} \|\mathbf{W}_l\|_2 \leqslant 0.5/L$, it holds that,

$$\|\nabla_{\mathbf{W}_l} L(\mathbf{W})\|_F^2 \geqslant \frac{2}{e} \sigma_{\min}^2(\mathbf{A}) \sigma_{\min}^2(\mathbf{B}) \sigma_r^2(\mathbf{X}) \big( L(\mathbf{W}) - L(\mathbf{W}^*) \big),$$

$$\|\nabla_{\mathbf{W}_l} L(\mathbf{W})\|_F^2 \leqslant 2e \|\mathbf{A}\|_2^2 \|\mathbf{B}\|_2^2 \|\mathbf{X}\|_2^2 \big( L(\mathbf{W}) - L(\mathbf{W}^*) \big)$$

$$\|\nabla_{\mathbf{W}_l} \ell(\mathbf{W}; \mathbf{x}_i, \mathbf{y}_i)\|_F^2 \leqslant 2e \|\mathbf{A}\|_2^2 \|\mathbf{B}\|_2^2 \|\mathbf{x}_i\|_2^2 \ell(\mathbf{W}; \mathbf{x}_i, \mathbf{y}_i).$$

In addition, the stochastic gradient $\mathbf{G}_l$ in Algorithm 1 satisfies

$$\|\mathbf{G}_l\|_F^2 \leqslant \frac{2en^2 \|\mathbf{A}\|_2^2 \|\mathbf{B}\|_2^2 \|\mathbf{X}\|_2^2}{B^2} L(\mathbf{W}),$$

where $B$ is the minibatch size.

The gradient lower bound can be also interpreted as the Polyak-Łojasiewicz condition, which is essential to the linear convergence rate. The gradient upper bound is crucial to bound the trajectory length, since this lemma requires that $\max_{l \in [L]} \|\mathbf{W}_l\| \leqslant 0.5/L$.

The following lemma proves the smoothness property of the training loss function $L(\mathbf{W})$ when $\mathbf{W}$ is staying inside a certain region. Its proof is in Section B.2.

**Lemma A.2.** For any two collections of weight matrices, denoted by $\widetilde{\mathbf{W}} = \{\widetilde{\mathbf{W}}_1, \ldots, \widetilde{\mathbf{W}}_L\}$ and $\mathbf{W} = \{\mathbf{W}_1, \ldots, \mathbf{W}_L\}$, satisfying $\max_{l \in [L]} \|\mathbf{W}_l\|_F, \max_{l \in [L]} \|\widetilde{\mathbf{W}}_l\|_F \leqslant 0.5/L$ that, it holds that

$$L(\widetilde{\mathbf{W}}) - L(\mathbf{W}) \leqslant \sum_{l=1}^{L} \langle \nabla_{\mathbf{W}_l} L(\mathbf{W}), \widetilde{\mathbf{W}}_l - \mathbf{W}_l \rangle$$

$$+ L \|\mathbf{A}\|_2 \|\mathbf{B}\|_2 \|\mathbf{X}\|_2 \big( \sqrt{2eL(\mathbf{W})} + 0.5e \|\mathbf{A}\|_2 \|\mathbf{B}\|_2 \|\mathbf{X}\|_2 \big) \sum_{l=1}^{L} \|\widetilde{\mathbf{W}}_l - \mathbf{W}_l\|_F^2.$$

Based on these two lemmas, we are able to complete the proof of all theorems, which are provided as follows.

## A.1 PROOF OF THEOREM 3.1

*Proof of Theorem 3.1.* In order to simplify the proof, we use the short-hand notations $\lambda_A$, $\mu_A$, $\lambda_B$ and $\mu_B$ to denote $\|\mathbf{A}\|_2$, $\sigma_{\min}(\mathbf{A})$, $\|\mathbf{B}\|_2$ and $\sigma_{\min}(\mathbf{B})$ respectively. Specifically, we rewrite the condition on $\mathbf{A}$ and $\mathbf{B}$ as follows

$$\frac{\mu_A^2 \mu_B^2}{\lambda_A \lambda_B} \geqslant \frac{4\sqrt{2e^3} \|\mathbf{X}\|_2}{\sigma_r^2(\mathbf{X})} \cdot \big( L(\mathbf{W}^{(0)}) - L(\mathbf{W}^*) \big)^{1/2}.$$

We prove the theorem by induction on the update number $s$, using the following two-part inductive hypothesis:

(i) $\max_{l \in [L]} \|\mathbf{W}_l^{(s)}\|_F \leqslant 0.5/L$,

(ii) $L(\mathbf{W}^{(s)}) - L(\mathbf{W}^*) \leqslant \left( 1 - \frac{\eta L \mu_A^2 \mu_B^2 \sigma_r^2(\mathbf{X})}{e} \right)^s \cdot \big( L(\mathbf{W}^{(0)}) - L(\mathbf{W}^*) \big).$

First, it can be easily verified that this holds for $s = 0$. Now, assume that the inductive hypothesis holds for $s < t$.

**Induction for Part (i):** We first prove that $\max_{l \in [L]} \|\mathbf{W}_l^{(t)}\|_F \leqslant 0.5/L$. By triangle inequality and the update rule of gradient descent, we have

$$\|\mathbf{W}_l^{(t)}\|_F \leqslant \sum_{s=0}^{t-1} \eta \|\nabla_{\mathbf{W}_l} L(\mathbf{W}^{(s)})\|_F$$

$$\leqslant \eta \sum_{s=0}^{t-1} \sqrt{2e} \lambda_A \lambda_B \|\mathbf{X}\|_2 \cdot \big( L(\mathbf{W}^{(s)}) - L(\mathbf{W}^*) \big)^{1/2}$$

$$\leqslant \sqrt{2e} \eta \lambda_A \lambda_B \|\mathbf{X}\|_2 \cdot \big( L(\mathbf{W}^{(0)}) - L(\mathbf{W}^*) \big)^{1/2} \cdot \sum_{s=0}^{t-1} \left( 1 - \frac{\eta L \mu_A^2 \mu_B^2 \sigma_r^2(\mathbf{X})}{e} \right)^{s/2}$$

where the second inequality follows from Lemma A.1, and the third inequality follows from the inductive hypothesis. Since $\sqrt{1-x} \leqslant 1 - x/2$ for any $x \in [0,1]$, we further have

$$\|\mathbf{W}_l^{(t)}\|_F \leqslant \sqrt{2e}\eta\lambda_A\lambda_B\|\mathbf{X}\|_2 \cdot \left(L(\mathbf{W}^{(0)}) - L(\mathbf{W}^*)\right)^{1/2} \cdot \sum_{s=0}^{t-1}\left(1 - \frac{\eta L\mu_A^2\mu_B^2\sigma_r^2(\mathbf{X})}{2e}\right)^s$$

$$\leqslant \frac{\sqrt{8e^3}\lambda_A\lambda_B\|\mathbf{X}\|_2}{L\mu_A^2\mu_B^2\sigma_r^2(\mathbf{X})} \cdot \left(L(\mathbf{W}^{(0)}) - L(\mathbf{W}^*)\right)^{1/2}.$$

Under the condition that $\mu_A^2\mu_B^2/(\lambda_A\lambda_B) \geqslant 2\sqrt{8e^3}\|\mathbf{X}\|_2\left(L(\mathbf{W}^{(0)}) - L(\mathbf{W}^*)\right)^{1/2}/\sigma_r^2(\mathbf{X})$, it can be readily verified that $\|\mathbf{W}_l^{(t)}\|_F \leqslant 0.5/L$. Since this holds for all $l \in [L]$, we have proved Part (i) of the inductive step, i.e., $\max_{l\in[L]}\|\mathbf{W}_l^{(t)}\|_F \leqslant 0.5/L$.

**Induction for Part (ii):** Now we prove Part (ii) of the inductive step, bounding the improvement in the objective function. Note that we have already shown that $\mathbf{W}^{(t)}$ satisfies $\max_{l\in[L]}\|\mathbf{W}_l^{(t)}\|_F \leqslant 0.5/L$, thus by Lemma A.2 we have

$$L(\mathbf{W}^{(t)}) \leqslant L(\mathbf{W}^{(t-1)}) - \eta\sum_{l=1}^{L}\left\|\nabla_{\mathbf{W}_l}L(\mathbf{W}^{(t-1)})\right\|_F^2$$

$$+ \eta^2 L\lambda_A\lambda_B\|\mathbf{X}\|_2 \cdot \left(\sqrt{eL(\mathbf{W}^{(t-1)})} + 0.5e\lambda_A\lambda_B\|\mathbf{X}\|_2\right) \cdot \sum_{l=1}^{L}\left\|\nabla_{\mathbf{W}_l}L(\mathbf{W}^{(t-1)})\right\|_F^2,$$

where we use the fact that $\mathbf{W}_l^{(t)} - \mathbf{W}_l^{(t-1)} = -\eta\nabla_{\mathbf{W}_l}L(\mathbf{W}^{(l-1)})$. Note that $L(\mathbf{W}^{(t-1)}) \leqslant L(\mathbf{W}^{(0)})$ and the step size is set to be

$$\eta = \frac{1}{2L\lambda_A\lambda_B\|\mathbf{X}\|_2 \cdot \left(\sqrt{eL(\mathbf{W}^{(0)})} + 0.5e\lambda_A\lambda_B\|\mathbf{X}\|_2\right)},$$

so that we have

$$L(\mathbf{W}^{(t)}) - L(\mathbf{W}^{(t-1)}) \leqslant -\frac{\eta}{2}\sum_{l=1}^{L}\left\|\nabla_{\mathbf{W}_l}L(\mathbf{W}^{(t-1)})\right\|_F^2$$

$$\leqslant -\frac{\eta L\mu_A^2\mu_B^2\sigma_r^2(\mathbf{X})}{e}\left(L(\mathbf{W}^{(t-1)}) - L(\mathbf{W}^*)\right),$$

where the second inequality is by Lemma A.1. Applying the inductive hypothesis, we get

$$L(\mathbf{W}^{(t)}) - L(\mathbf{W}^*) \leqslant \left(1 - \frac{\eta L\mu_A^2\mu_B^2\sigma_r^2(\mathbf{X})}{e}\right) \cdot \left(L(\mathbf{W}^{(t-1)}) - L(\mathbf{W}^*)\right)$$

$$\leqslant \left(1 - \frac{\eta L\mu_A^2\mu_B^2\sigma_r^2(\mathbf{X})}{e}\right)^t \cdot \left(L(\mathbf{W}^{(0)}) - L(\mathbf{W}^*)\right), \tag{A.1}$$

which completes the proof of the inductive step of Part (ii). Thus we are able to complete the proof. $\square$

## A.2 PROOF OF PROPOSITION 3.3

*Proof of Proposition 3.3.* We prove the bounds on the singular values and initial training loss separately.

**Bounds on the singular values:** Specifically, we set the neural network width as

$$m \geqslant 100 \cdot \left(\sqrt{\max\{d,k\}} + \sqrt{2\log(12/\delta)}\right)^2$$

By Corollary 5.35 in Vershynin (2010), we know that for a matrix $\mathbf{U} \in \mathbb{R}^{d_1 \times d_2}$ ($d_1 \geqslant d_2$) with entries independently generated by standard normal distribution, with probability at least $1 - 2\exp(-t^2/2)$, its singular values satisfy

$$\sqrt{d_1} - \sqrt{d_2} - t \leqslant \sigma_{\min}(\mathbf{U}) \leqslant \sigma_{\max}(\mathbf{U}) \leqslant \sqrt{d_1} + \sqrt{d_2} + t.$$

Based on our constructions of $\mathbf{A}$ and $\mathbf{B}$, we know that each entry of $\frac{1}{\beta}\mathbf{B}$ and $\frac{1}{\alpha}\mathbf{A}$ follows standard Gaussian distribution. Therefore, set $t = 2\sqrt{\log(12/\delta)}$ and apply union bound, with probability at least $1 - \delta/3$, the following holds,

$$\alpha\big(\sqrt{m} - \sqrt{d} - 2\sqrt{\log(12/\delta)}\big) \leqslant \sigma_{\min}(\mathbf{A}) \leqslant \sigma_{\max}(\mathbf{A}) \leqslant \alpha\big(\sqrt{m} + \sqrt{d} + 2\sqrt{\log(12/\delta)}\big)$$
$$\beta\big(\sqrt{m} - \sqrt{k} - 2\sqrt{\log(12/\delta)}\big) \leqslant \sigma_{\min}(\mathbf{B}) \leqslant \sigma_{\max}(\mathbf{B}) \leqslant \beta\big(\sqrt{m} + \sqrt{k} + 2\sqrt{\log(12/\delta)}\big),$$

where we use the facts that $\sigma_{\min}(\kappa\mathbf{U}) = \kappa\sigma_{\min}(\mathbf{U})$ and $\sigma_{\max}(\kappa\mathbf{U}) = \kappa\sigma_{\max}(\mathbf{U})$ for any scalar $\kappa$ and matrix $\mathbf{U}$. Then applying our choice of $m$, we have with probability at least $1 - \delta/3$,

$$0.9\alpha\sqrt{m} \leqslant \sigma_{\min}(\mathbf{A}) \leqslant \sigma_{\max}(\mathbf{A}) \leqslant 1.1\alpha\sqrt{m} \quad \text{and} \quad 0.9\beta\sqrt{m} \leqslant \sigma_{\min}(\mathbf{B}) \leqslant \sigma_{\max}(\mathbf{B}) \leqslant 1.1\beta\sqrt{m}.$$

This completes the proof of the bounds on the singular values of $\mathbf{A}$ and $\mathbf{B}$.

**Bounds on the initial training loss:** The proof in this part is similar to the proof of Proposition 6.5 in Du & Hu (2019). Since we apply zero initialization on all hidden layers, by Young's inequality, we have the following for any $(\mathbf{x}, \mathbf{y})$,

$$\ell(\mathbf{W}^{(0)}; \mathbf{x}, \mathbf{y}) = \frac{1}{2}\|\mathbf{B}\mathbf{A}\mathbf{x} - \mathbf{y}\|_2^2 \leqslant \|\mathbf{B}\mathbf{A}\mathbf{x}\|_2^2 + \|\mathbf{y}\|_2^2. \tag{A.2}$$

Since each entry of $\mathbf{B}$ is generated from $\mathcal{N}(0, \beta^2)$, conditioned on $\mathbf{A}$, each entry of $\mathbf{B}\mathbf{A}\mathbf{x}$ is distributed according to $\mathcal{N}(0, \beta^2\|\mathbf{A}\mathbf{x}\|_2^2)$, so $\frac{\|\mathbf{B}\mathbf{A}\mathbf{x}\|_2^2}{\|\mathbf{A}\mathbf{x}\|_2^2\beta^2}$ follows a $\chi_k^2$ distribution. Applying a standard tail bound for $\chi_k^2$ distribution, we have, with probability at least $1 - \delta'$,

$$\frac{\|\mathbf{B}\mathbf{A}\mathbf{x}\|_2^2}{\|\mathbf{A}\mathbf{x}\|_2^2} \leqslant \beta^2 k(1 + 2\sqrt{\log(1/\delta')/k} + 2\log(1/\delta')).$$

Note that by our bounds of the singular values, if $m \geqslant 100 \cdot \big(\sqrt{\max\{d, k\}} + \sqrt{2\log(8/\delta)}\big)^2$, we have with probability at least $1 - \delta/3$, $\|\mathbf{A}\|_2 \leqslant 1.1\alpha\sqrt{m}$, thus, it follows that with probability at least $1 - \delta' - \delta$,

$$\|\mathbf{B}\mathbf{A}\mathbf{x}\|_2^2 \leqslant 1.21\alpha^2\beta^2 km\big[1 + 2\sqrt{\log(1/\delta')} + 2\log(1/\delta')\big]\|\mathbf{x}\|_2^2.$$

Then by union bound, it is evident that with probability $1 - n\delta' - \delta/3$,

$$\|\mathbf{B}\mathbf{A}\mathbf{X}\|_F^2 = \sum_{i=1}^{n} \|\mathbf{B}\mathbf{A}\mathbf{x}_i\|_2^2 \leqslant 1.21\alpha^2\beta^2 km\big[1 + 2\sqrt{\log(1/\delta')} + 2\log(1/\delta')\big]\|\mathbf{X}\|_F^2.$$

Set $\delta' = \delta/(3n)$, suppose $\log(1/\delta') \geqslant 1$, we have with probability at least $1 - 2\delta/3$,

$$L(\mathbf{W}^{(0)}) = \frac{1}{2}\|\mathbf{B}\mathbf{A}\mathbf{X} - \mathbf{Y}\|_F^2 \leqslant \|\mathbf{B}\mathbf{A}\mathbf{X}\|_F^2 + \|\mathbf{Y}\|_F^2 \leqslant 6.05\alpha^2\beta^2 km\log(2n/\delta)\|\mathbf{X}\|_F^2 + \|\mathbf{Y}\|_F^2.$$

This completes the proof of the bounds on the initial training loss.

Applying a union bound on these two parts, we are able to complete the proof. □

## A.3 PROOF OF COROLLARY 3.4

*Proof of Corollary 3.4.* Recall the condition in Theorem 3.1:

$$\frac{\sigma_{\min}^2(\mathbf{A})\sigma_{\min}^2(\mathbf{B})}{\|\mathbf{A}\|_2\|\mathbf{B}\|_2} \geqslant C \cdot \frac{\|\mathbf{X}\|_2}{\sigma_r^2(\mathbf{X})} \cdot \big(L(\mathbf{W}^{(0)}) - L(\mathbf{W}^*)\big)^{1/2}. \tag{A.3}$$

By Proposition 3.3, we know that, with probability $1 - \delta$,

$$\frac{\sigma_{\min}^2(\mathbf{A})\sigma_{\min}^2(\mathbf{B})}{\|\mathbf{A}\|_2\|\mathbf{B}\|_2} = \Theta(m),$$

$$\frac{\|\mathbf{X}\|_2}{\sigma_r(\mathbf{X})} \cdot \big(L(\mathbf{W}^{(0)}) - L(\mathbf{W}^*)\big)^{1/2} = O\left(\frac{(\sqrt{km\log(n/\delta)} + 1)\|\mathbf{X}\|_F\|\mathbf{X}\|_2}{\sigma_r(\mathbf{X})}\right).$$

Note that $\|\mathbf{X}\|_F \leqslant \sqrt{r}\|\mathbf{X}\|_2$, thus the condition (A.3) can be satisfied if $m = \Omega(kr\kappa^2 \log(n/\delta))$ where $\kappa = \|\mathbf{X}\|_2^2/\sigma_r^2(\mathbf{X})$.

Theorem 3.1 implies that $L(\mathbf{W}^{(t)}) - L(\mathbf{W}^*) \leqslant \epsilon$ after $T = O\left(\frac{1}{\eta L \sigma_{\min}^2(\mathbf{A})\sigma_{\min}^2(\mathbf{B})\sigma_r^2(\mathbf{X})} \log \frac{1}{\epsilon}\right)$ iterations. Plugging in the value of $\eta$, we get

$$T = O\left(\frac{\|\mathbf{A}\|_2\|\mathbf{B}\|_2\|\mathbf{X}\|_2 \cdot \left(\sqrt{L(\mathbf{W}^{(0)})} + \|\mathbf{A}\|_2\|\mathbf{B}\|_2\|\mathbf{X}\|_2\right)}{\sigma_{\min}^2(\mathbf{A})\sigma_{\min}^2(\mathbf{B})\sigma_r^2(\mathbf{X})} \log \frac{1}{\epsilon}\right).$$

By Proposition 3.3, we have

$$T = O\left(\frac{\|\mathbf{A}\|_2\|\mathbf{B}\|_2\|\mathbf{X}\|_2 \cdot \left(\sqrt{km\log(n/\delta)}\|\mathbf{X}\|_F + \|\mathbf{A}\|_2\|\mathbf{B}\|_2\|\mathbf{X}\|_2\right)}{\sigma_{\min}^2(\mathbf{A})\sigma_{\min}^2(\mathbf{B})\sigma_r^2(\mathbf{X})} \log \frac{1}{\epsilon}\right)$$

$$= O\left(\frac{\|\mathbf{X}\|_2 \cdot \left(\sqrt{km\log(n/\delta)}\|\mathbf{X}\|_F + m\|\mathbf{X}\|_2\right)}{m\sigma_r^2(\mathbf{X})} \log \frac{1}{\epsilon}\right)$$

$$= O\left(\frac{\|\mathbf{X}\|_2 \cdot \left(\sqrt{kr\log(n/\delta)/m}\|\mathbf{X}\|_2 + \|\mathbf{X}\|_2\right)}{\sigma_r^2(\mathbf{X})} \log \frac{1}{\epsilon}\right)$$

$$= O\left(\kappa \log \frac{1}{\epsilon}\right)$$

for $m = \Omega(kr\log(n/\delta))$, completing the proof. $\qquad\square$

## A.4 PROOF OF THEOREM 3.6

*Proof of Theorem 3.6.* The guarantee is already achieved by $\mathbf{W}^{(0)}$ if $\epsilon \geqslant L(\mathbf{W}^{(0)}) - L(\mathbf{W}^*)$, so we may assume without loss of generality that $\epsilon < L(\mathbf{W}^{(0)}) - L(\mathbf{W}^*)$.

Similar to the proof of Theorem 3.1, we use the short-hand notations $\lambda_A$, $\mu_A$, $\lambda_B$ and $\mu_B$ to denote $\|\mathbf{A}\|_2$, $\sigma_{\min}(\mathbf{A})$, $\|\mathbf{B}\|_2$ and $\sigma_{\min}(\mathbf{B})$ respectively. Then we rewrite the condition on $\mathbf{A}$ and $\mathbf{B}$, and our choices of $\eta$ and $T$ as follows

$$\frac{\mu_A^2\mu_B^2}{\lambda_A\lambda_B} \geqslant \frac{\sqrt{8e^3}n\|\mathbf{X}\|_2 \cdot \log(L(\mathbf{W}^{(0)})/\epsilon')}{B\sigma_r^2(\mathbf{X})} \cdot \sqrt{2L(\mathbf{W}^{(0)})}$$

$$\eta \leqslant \frac{B\mu_A^2\mu_B^2\sigma_r^2(\mathbf{X})}{6e^3 Ln\lambda_A^4\lambda_B^4\|\mathbf{X}\|_2^2} \cdot \min\left\{\frac{\epsilon'}{\|\mathbf{X}\|_{2,\infty}^2 L(\mathbf{W}^*)}, \frac{\log^2(2)B}{3n\|\mathbf{X}\|_2^2 \cdot \log(T/\delta)\log(L(\mathbf{W}^{(0)})/\epsilon')}\right\},$$

$$T = \frac{e}{\eta L\mu_A^2\mu_B^2\sigma_r^2(\mathbf{X})} \cdot \log\left(\frac{L(\mathbf{W}^{(0)}) - L(\mathbf{W}^*)}{\epsilon'}\right),$$

where we set $\epsilon' = \epsilon/3$ for the purpose of the proof.

We first prove the convergence guarantees on expectation, and then apply the Markov inequality.

For SGD, our guarantee is not made on the last iterate but the best one. Define $\mathfrak{E}_t$ to be the event that there is no $s \leqslant t$ such that $L(\mathbf{W}^{(t)}) - L(\mathbf{W}^*) \leqslant \epsilon'$. If $\mathbb{1}(\mathfrak{E}_t) = 0$, then there is an iterate $\mathbf{W}_s$ with $s \leqslant t$ that achieves training loss within $\epsilon'$ of optimal.

Similar to the proof of Theorem 3.1, we prove the theorem by induction on the update number $s$, using the following inductive hypothesis: either $\mathbb{1}(\mathfrak{E}_s) = 0$ or the following three inequalities hold,

(i) $\max_{l\in[L]} \|\mathbf{W}_l^{(s)}\|_F \leqslant \frac{\sqrt{2e}s\eta n\lambda_A\lambda_B\|\mathbf{X}\|_2}{B} \cdot \sqrt{2L(\mathbf{W}^{(0)})}.$

(ii) $\mathbb{E}\left[\left(L(\mathbf{W}^{(s)}) - L(\mathbf{W}^*)\right)\right] \leqslant \left(1 - \frac{\eta L\mu_A^2\mu_B^2\sigma_r^2(\mathbf{X})}{e}\right)^s \cdot \left(L(\mathbf{W}^{(0)}) - L(\mathbf{W}^*)\right)$

(iii) $L(\mathbf{W}^{(s)}) \leqslant 2L(\mathbf{W}^{(0)}),$

where the expectation in Part (ii) is with respect to all of the random choices of minibatches. Clearly, if $\mathbb{1}(\mathfrak{E}_s) = 0$, we have already finished the proof since there is an iterate that achieves training loss

within $\epsilon'$ of optimal. Recalling that $\epsilon < L(\mathbf{W}^{(0)}) - L(\mathbf{W}^*)$, it is easy to verify that the inductive hypothesis holds when $s = 0$.

For the inductive step, we will prove that if the inductive hypothesis holds for $s < t$, then it holds for $s = t$. When $\mathbb{1}(\mathfrak{E}_{t-1}) = 0$, then $\mathbb{1}(\mathfrak{E}_t)$ is also 0 and we are done. Therefore, the remaining part is to prove the inductive hypothesis for $s = t$ under the assumption that $\mathbb{1}(\mathfrak{E}_{t-1}) = 1$, which implies that (i), (ii) and (iii) hold for all $s \leqslant t - 1$. For Parts (i) and (ii), we will directly prove that the corresponding two inequalities hold. For Part (iii), we will prove that either this inequality holds or $\mathbb{1}(\mathfrak{E}_t) = 0$.

**Induction for Part (i):** As we mentioned, this part will be proved under the assumption $\mathbb{1}(\mathfrak{E}_{t-1}) = 1$. Besides, combining Part (i) for $s = t - 1$ and our choice of $\eta$ and $T$ implies that $\max_{l \in [L]} \|\mathbf{W}_l^{(t-1)}\|_F \leqslant 0.5/L$. Then by triangle inequality, we have the following for $\|\mathbf{W}_l^{(t)}\|_F$,

$$\|\mathbf{W}_l^{(t)}\|_F \leqslant \|\mathbf{W}_l^{(t-1)}\|_F + \eta \|\mathbf{G}_l^{(t-1)}\|_F.$$

By Lemma A.1, we have

$$\|\mathbf{G}_l^{(t-1)}\|_F \leqslant \frac{\sqrt{2e}n\lambda_A\lambda_B\|\mathbf{X}\|_2}{B} \cdot \sqrt{L(\mathbf{W}^{(t-1)})}.$$

Then we have

$$
\|\mathbf{W}_l^{(t)}\|_F \leqslant \left(\|\mathbf{W}_l^{(t-1)}\|_F + \eta\|\mathbf{G}_l^{(t-1)}\|_F\right)
$$
$$
\leqslant \|\mathbf{W}_l^{(t-1)}\|_F + \frac{\sqrt{2e}\eta n\lambda_A\lambda_B\|\mathbf{X}\|_2}{B} \cdot \sqrt{L(\mathbf{W}^{(t-1)})}. \tag{A.4}
$$

By Part (iii) for $s = t - 1$, we know that $L(\mathbf{W}^{(t-1)}) \leqslant 2L(\mathbf{W}^{(0)})$. Then by Part (i) for $s = t - 1$, it is evident that

$$\|\mathbf{W}_l^{(t)}\|_F \leqslant \frac{\sqrt{2e}t\eta n\lambda_A\lambda_B\|\mathbf{X}\|_2}{B} \cdot \sqrt{2L(\mathbf{W}^{(0)})}. \tag{A.5}$$

This completes the proof of the inductive step of Part (i).

**Induction for Part (ii):** As we previously mentioned, we will prove this part under the assumption $\mathbb{1}(\mathfrak{E}_{t-1}) = 1$. Thus, as mentioned earlier, the inductive hypothesis implies that $\max_{l \in [L]} \|\mathbf{W}_l^{(t-1)}\|_F \leqslant 0.5/L$. By Part (i) for $s = t$, which has been verified in (A.5), it can be proved that $\max_{l \in [L]} \|\mathbf{W}_l^{(t)}\|_F \leqslant 0.5/L$, then we have the following by Lemma A.2,

$$
L(\mathbf{W}^{(t)}) - L(\mathbf{W}^{(t-1)}) \leqslant -\eta \sum_{l=1}^{L} \left\langle \nabla_{\mathbf{W}_l} L(\mathbf{W}^{(t-1)}), \mathbf{G}_l^{(t-1)} \right\rangle
$$
$$
+ \eta^2 L\lambda_A\lambda_B\|\mathbf{X}\|_2 \cdot \left(\sqrt{eL(\mathbf{W}^{(t-1)})} + 0.5e\lambda_A\lambda_B\|\mathbf{X}\|_2\right) \cdot \sum_{l=1}^{L} \|\mathbf{G}_l^{(t-1)}\|_F^2. \tag{A.6}
$$

By our condition on $\mathbf{A}$ and $\mathbf{B}$, it is easy to verify that

$$\lambda_A\lambda_B \geqslant \frac{\mu_A^2\mu_B^2}{\lambda_A\lambda_B} \geqslant \frac{2\sqrt{2e^{-1}L(\mathbf{W}^{(0)})}}{\|\mathbf{X}\|_2}.$$

Then by Part (iii) for $s = t - 1$ (A.6) yields

$$
L(\mathbf{W}^{(t)}) - L(\mathbf{W}^{(t-1)}) \leqslant -\eta \sum_{l=1}^{L} \left\langle \nabla_{\mathbf{W}_l} L(\mathbf{W}^{(t-1)}), \mathbf{G}_l^{(t-1)} \right\rangle + e\eta^2 L\lambda_A^2\lambda_B^2\|\mathbf{X}\|_2^2 \cdot \sum_{l=1}^{L} \|\mathbf{G}_l^{(t-1)}\|_F^2. \tag{A.7}
$$

Taking expectation conditioning on $\mathbf{W}^{(t-1)}$ gives

$$
\mathbb{E}\left[L(\mathbf{W}^{(t)})|\mathbf{W}^{(t-1)}\right] - L(\mathbf{W}^{(t-1)}) \leqslant -\eta \sum_{l=1}^{L} \left\|\nabla_{\mathbf{W}_l} L(\mathbf{W}^{(t-1)})\right\|_F^2
$$
$$
+ e\eta^2 L\lambda_A^2\lambda_B^2\|\mathbf{X}\|_2^2 \sum_{l=1}^{L} \mathbb{E}\left[\|\mathbf{G}_l^{(t-1)}\|_F^2|\mathbf{W}^{(t-1)}\right]. \tag{A.8}
$$

Note that, for $i$ sampled uniformly from $\{1, ..., n\}$, the expectation $\mathbb{E}[\|\mathbf{G}_l^{(t-1)}\|_F^2|\mathbf{W}^{(t-1)}]$ can be upper bounded by

$$
\begin{aligned}
\mathbb{E}[\|\mathbf{G}_l^{(t-1)}\|_F^2|\mathbf{W}^{(t-1)}] &= \mathbb{E}[\|\mathbf{G}_l^{(t-1)} - \nabla_{\mathbf{W}_l}L(\mathbf{W}^{(t-1)})\|_F^2|\mathbf{W}^{(t-1)}] + \|\nabla_{\mathbf{W}_l}L(\mathbf{W}^{(t-1)})\|_F^2 \\
&\leqslant \frac{n^2}{B}\mathbb{E}[\|\nabla_{\mathbf{W}_l}\ell(\mathbf{W}^{(t-1)};\mathbf{x}_i,\mathbf{y}_i)\|_F^2|\mathbf{W}^{(t-1)}] + \|\nabla_{\mathbf{W}_l}L(\mathbf{W}^{(t-1)})\|_F^2.
\end{aligned}
$$
(A.9)

By Lemma A.1, we have

$$
\begin{aligned}
\mathbb{E}[\|\nabla_{\mathbf{W}_l}\ell(\mathbf{W}^{(t-1)};\mathbf{x}_i,\mathbf{y}_i)\|_F^2|\mathbf{W}^{(t-1)}] &\leqslant 2e\lambda_A^2\lambda_B^2\mathbb{E}[\|\mathbf{x}_i\|_2^2\ell(\mathbf{W}^{(t-1)};\mathbf{x}_i,\mathbf{y}_i)|\mathbf{W}^{(t-1)}] \\
&\leqslant \frac{2e\lambda_A^2\lambda_B^2}{n}\sum_{i=1}^{n}\|\mathbf{x}_i\|_2^2\ell(\mathbf{W}^{(t-1)};\mathbf{x}_i,\mathbf{y}_i) \\
&\leqslant \frac{2e\lambda_A^2\lambda_B^2\|\mathbf{X}\|_{2,\infty}^2 L(\mathbf{W}^{(t-1)})}{n}.
\end{aligned}
$$

Plugging the above inequality into (A.9) and (A.8), we get

$$
\begin{aligned}
&\mathbb{E}[L(\mathbf{W}^{(t)})|\mathbf{W}^{(t-1)}] - L(\mathbf{W}^{(t-1)}) \\
&\leqslant -\eta\sum_{l=1}^{L}\|\nabla_{\mathbf{W}_l}L(\mathbf{W}^{(t-1)})\|_F^2 \\
&\quad + e\eta^2 L\lambda_A^2\lambda_B^2\|\mathbf{X}\|_2^2 \cdot \sum_{l=1}^{L}\left(\frac{2en\lambda_A^2\lambda_B^2\|\mathbf{X}\|_{2,\infty}^2 L(\mathbf{W}^{(t-1)})}{B} + \|\nabla_{\mathbf{W}_l}L(\mathbf{W}^{(t-1)})\|_F^2\right).
\end{aligned}
$$

Recalling that $\eta \leqslant 1/(6eL\lambda_A^2\lambda_B^2\|\mathbf{X}\|_2^2)$, we have

$$
\begin{aligned}
\mathbb{E}[L(\mathbf{W}^{(t)})|\mathbf{W}^{(t-1)}] - L(\mathbf{W}^{(t-1)}) &\leqslant -\frac{5\eta}{6}\sum_{l=1}^{L}\|\nabla_{\mathbf{W}_l}L(\mathbf{W}^{(t-1)})\|_F^2 \\
&\quad + \frac{2e^2\eta^2 L^2 n\lambda_A^4\lambda_B^4\|\mathbf{X}\|_2^2\|\mathbf{X}\|_{2,\infty}^2 L(\mathbf{W}^{(t-1)})}{B}.
\end{aligned}
$$
(A.10)

By Lemma A.1, we have

$$
\sum_{l=1}^{L}\|\nabla_{\mathbf{W}_l}L(\mathbf{W}^{(t-1)})\|_F^2 \geqslant 2e^{-1}L\mu_A^2\mu_B^2\sigma_r^2(\mathbf{X})\left(L(\mathbf{W}^{(t-1)}) - L(\mathbf{W}^*)\right).
$$

If we set

$$
\eta \leqslant \frac{B\mu_A^2\mu_B^2\sigma_r^2(\mathbf{X})}{6e^3 Ln\lambda_A^4\lambda_B^4\|\mathbf{X}\|_2^2\|\mathbf{X}\|_{2,\infty}^2},
$$
(A.11)

then (A.10) yields

$$
\begin{aligned}
&\mathbb{E}[L(\mathbf{W}^{(t)})|\mathbf{W}^{(t-1)}] - L(\mathbf{W}^{(t-1)}) \\
&\leqslant -\frac{5\eta L\mu_A^2\mu_B^2\sigma_r^2(\mathbf{X})}{3e}\left(L(\mathbf{W}^{(t-1)}) - L(\mathbf{W}^*)\right) \\
&\quad + \frac{2e^2\eta^2 L^2 n\lambda_A^4\lambda_B^4\|\mathbf{X}\|_2^2\|\mathbf{X}\|_{2,\infty}^2\left(L(\mathbf{W}^{(t-1)}) - L(\mathbf{W}^*)\right)}{B} \\
&\quad + \frac{2e^2\eta^2 L^2 n\lambda_A^4\lambda_B^4\|\mathbf{X}\|_2^2\|\mathbf{X}\|_{2,\infty}^2 L(\mathbf{W}^*)}{B} \\
&\leqslant -\frac{4\eta L\mu_A^2\mu_B^2\sigma_r^2(\mathbf{X})}{3e}\left(L(\mathbf{W}^{(t-1)}) - L(\mathbf{W}^*)\right) + \frac{2e^2\eta^2 L^2 n\lambda_A^4\lambda_B^4\|\mathbf{X}\|_2^2\|\mathbf{X}\|_{2,\infty}^2 L(\mathbf{W}^*)}{BL^2}.
\end{aligned}
$$
(A.12)

Define

$$
\gamma_0 = \frac{4L\mu_A^2\mu_B^2\sigma_r^2(\mathbf{X})}{3e}, \quad \text{and} \quad \gamma_1 = \frac{2e^2\eta^2 L^2 n\lambda_A^4\lambda_B^4\|\mathbf{X}\|_2^2\|\mathbf{X}\|_{2,\infty}^2 L(\mathbf{W}^*)}{B},
$$

rearranging (A.12) further gives

$$\mathbb{E}\big[L(\mathbf{W}^{(t)})|\mathbf{W}^{(t-1)}\big] - L(\mathbf{W}^*) \leqslant (1 - \eta\gamma_0) \cdot \big(L(\mathbf{W}^{(t)}) - L(\mathbf{W}^*)\big) + \eta^2\gamma_1. \qquad (A.13)$$

Therefore, setting the step size as

$$\eta \leqslant \frac{\gamma_0\epsilon'}{4\gamma_1} = \frac{B\mu_A^2\mu_B^2\sigma_r^2(\mathbf{X})}{6e^3Ln\lambda_A^4\lambda_B^4\|\mathbf{X}\|_2^2\|\mathbf{X}\|_{2,\infty}^2} \cdot \frac{\epsilon'}{L(\mathbf{W}^*)},$$

we further have

$$\mathbb{E}\big[L(\mathbf{W}^{(t)}) - L(\mathbf{W}^*)|\mathbf{W}^{(t-1)}\big] \leqslant \big[(1 - \eta\gamma_0) \cdot [L(\mathbf{W}^{(t-1)}) - L(\mathbf{W}^*)] + \eta^2\gamma_1\big]$$
$$\leqslant (1 - 3\eta\gamma_0/4) \cdot [L(\mathbf{W}^{(t-1)}) - L(\mathbf{W}^*)], \qquad (A.14)$$

where the second inequality is by (A.13) and the last inequality is by the fact that we assume $\mathbb{1}(\mathfrak{E}_{t-1}) = 1$, which implies that $L(\mathbf{W}^{(t-1)}) - L(\mathbf{W}^*) \geqslant \epsilon' \geqslant 4\gamma_1\eta/\gamma_0$. Further taking expectation over $\mathbf{W}^{(t-1)}$, we get

$$\mathbb{E}\big[L(\mathbf{W}^{(t)}) - L(\mathbf{W}^*)\big] \leqslant (1 - 3\eta\gamma_0/4) \cdot \mathbb{E}\big[L(\mathbf{W}^{(t-1)}) - L(\mathbf{W}^*)\big]$$
$$\leqslant (1 - 3\eta\gamma_0/4)^t \cdot \big(L(\mathbf{W}^{(0)}) - L(\mathbf{W}^*)\big),$$

where the second inequality follows from Part (ii) for $s = t - 1$ and the assumption that $\mathbb{1}(\mathfrak{E}_0) = 1$. Plugging the definition of $\gamma_0$, we are able to complete the proof of the inductive step of Part (ii).

**Induction for Part (iii):** Recalling that for this part, we are going to prove that either $L(\mathbf{W}^{(t)}) \leqslant 2L(\mathbf{W}^{(0)})$ or $\mathbb{1}(\mathfrak{E}_t) = 0$, which is equivalent to $L(\mathbf{W}^{(t)}) \cdot \mathbb{1}(\mathfrak{E}_t) \leqslant 2L(\mathbf{W}^{(0)})$ since $L(\mathbf{W}^{(0)})$ and $L(\mathbf{W}^{(t)})$ are both positive. We will prove this by martingale inequality. Let $\mathcal{F}_t = \sigma\{\mathbf{W}^{(0)}, \cdots, \mathbf{W}^{(t)}\}$ be a $\sigma$-algebra, and $\mathbb{F} = \{\mathcal{F}_t\}_{t \geqslant 1}$ be a filtration. We first prove that $\mathbb{E}[L(\mathbf{W}^{(t)})\mathbb{1}(\mathfrak{E}_t)|\mathcal{F}_{t-1}] \leqslant L(\mathbf{W}^{(t-1)})\mathbb{1}(\mathfrak{E}_{t-1})$. Apparently, this inequality holds when $\mathbb{1}(\mathfrak{E}_{t-1}) = 0$ since both sides will be zero. Then if $\mathbb{1}(\mathfrak{E}_{t-1}) = 1$, by (A.14) we have $\mathbb{E}[L(\mathbf{W}^{(t)})|\mathbf{W}^{(t-1)}] \leqslant L(\mathbf{W}^{(t-1)})$ since $L(\mathbf{W}^*)$ is the global minimum. Therefore,

$$\mathbb{E}[L(\mathbf{W}^{(t)})\mathbb{1}(\mathfrak{E}_t)|\mathcal{F}_{t-1}, \mathbf{W}^{(t-1)}, \mathbb{1}(\mathfrak{E}_{t-1}) = 1] \leqslant \mathbb{E}[L(\mathbf{W}^{(t)})|\mathcal{F}_{t-1}, \mathbb{1}(\mathfrak{E}_{t-1}) = 1]$$
$$\leqslant L(\mathbf{W}^{(t-1)}).$$

Combining these two cases, by Jensen's inequality, we further have

$$\mathbb{E}\big[\log\big(L(\mathbf{W}^{(t)})\mathbb{1}(\mathfrak{E}_t)\big)|\mathcal{F}_{t-1}\big] \leqslant \log\big(\mathbb{E}[L(\mathbf{W}^{(t)})\mathbb{1}(\mathfrak{E}_t)|\mathcal{F}_{t-1}]\big)$$
$$\leqslant \log\big(L(\mathbf{W}^{(t-1)})\mathbb{1}(\mathfrak{E}_{t-1})\big),$$

which implies that $\{\log\big(L(\mathbf{W}^{(t)}) \cdot \mathbb{1}(\mathfrak{E}_t)\big)\}_{t \geqslant 0}$ is a super-martingale. Then we will upper bound the martingale difference $\log\big(L(\mathbf{W}^{(t)}) \cdot \mathbb{1}(\mathfrak{E}_t)\big) - \log\big(L(\mathbf{W}^{(t-1)}) \cdot \mathbb{1}(\mathfrak{E}_{t-1})\big)$. Clearly this quantity would be zero if $\mathbb{1}(\mathfrak{E}_{t-1}) = 0$. Then if $\mathbb{1}(\mathfrak{E}_{t-1}) = 1$, by (A.7) we have

$$L(\mathbf{W}^{(t)}) \leqslant L(\mathbf{W}^{(t-1)}) + \eta\sum_{l=1}^{L}\|\nabla_{\mathbf{W}_l}L(\mathbf{W}^{(t-1)})\|_F\|\mathbf{G}_l^{(t-1)}\|_F + e\eta^2L\lambda_A^2\lambda_B^2\|\mathbf{X}\|_2^2\sum_{l=1}^{L}\|\mathbf{G}_l^{(t-1)}\|_F^2.$$

By Part (i) for $s = t - 1$, Lemma A.1, we further have

$$L(\mathbf{W}^{(t)}) \leqslant \left(1 + \frac{2e\eta Ln\lambda_A^2\lambda_B^2\|\mathbf{X}\|_2^2}{B} + \frac{2e^2n^2\eta^2L^2\lambda_A^4\lambda_B^4\|\mathbf{X}\|_2^4}{B^2}\right)L(\mathbf{W}^{(t-1)})$$
$$\leqslant \left(1 + \frac{3e\eta nL\lambda_A^2\lambda_B^2\|\mathbf{X}\|_2^2}{B}\right)L(\mathbf{W}^{(t-1)}), \qquad (A.15)$$

where the second inequality follows from the choice of $\eta$ that

$$\eta \leqslant \frac{B}{2enL\lambda_A^2\lambda_B^2\|\mathbf{X}\|_2^2}.$$

Using the fact that $\mathbb{1}(\mathfrak{E}_t) \leqslant 1$ and $\mathbb{1}(\mathfrak{E}_{t-1}) = 1$, we further have

$$\log\big(L(\mathbf{W}^{(t)}) \cdot \mathbb{1}(\mathfrak{E}_t)\big) \leqslant \log\big(L(\mathbf{W}^{(t-1)}) \cdot \mathbb{1}(\mathfrak{E}_{t-1})\big) + \frac{3e\eta Ln\lambda_A^2\lambda_B^2\|\mathbf{X}\|_2^2}{B},$$

which also holds for the case $\mathbb{1}(\mathfrak{E}_{t-1}) = 0$. Recall that $\{\log\left(L(\mathbf{W}^{(t)}) \cdot \mathbb{1}(\mathfrak{E}_t)\right)\}_{t \geq 0}$ is a super-martingale, thus by one-side Azuma's inequality, we have with probability at least $1 - \delta'$,

$$\log\left(L(\mathbf{W}^{(t)}) \cdot \mathbb{1}(\mathfrak{E}_t)\right) \leq \log\left(L(\mathbf{W}^{(0)})\right) + \frac{3e\eta Ln\lambda_A^2 \lambda_B^2 \|\mathbf{X}\|_2^2}{B} \cdot \sqrt{2t\log(1/\delta')}.$$

Setting $\delta' = \delta/T$, using the fact that $t \leq T$ and leveraging our choice of $T$ and $\eta$, we have with probability at least $1 - \delta/T$,

$$\sqrt{T}\eta = \frac{\log(2)B}{3e\sqrt{2\log(\delta/T)}Ln\lambda_A^2 \lambda_B^2 \|\mathbf{X}\|_2^2},$$

which implies that

$$L(\mathbf{W}^{(t)})\mathbb{1}(\mathfrak{E}_t) \leq \exp\left[\log\left(L(\mathbf{W}^{(0)})\right) + \log(2)\right] \leq 2L(\mathbf{W}^{(0)}). \tag{A.16}$$

This completes the proof of the inductive step of Part (iii).

Note that this result holds with probability at least $1 - \delta/T$. Thus applying union bound over all iterates $\{\mathbf{W}^{(t)}\}_{t=0,\dots,T}$ yields that all induction arguments hold for all $t \leq T$ with probability at least $1 - \delta$.

Moreover, plugging our choice of $T$ and $\eta$ into Part (ii) gives

$$\mathbb{E}\left[L(\mathbf{W}^{(t)}) - L(\mathbf{W}^*)\right] \leq \epsilon'.$$

By Markov inequality, we further have with probability at least $2/3$, it holds that $[L(\mathbf{W}^{(T)}) - L(\mathbf{W}^*)] \cdot \mathbb{1}(\mathfrak{E}_t) \leq 3\epsilon' = \epsilon$. Therefore, by union bound (together with the high probability arguments of (A.16)) and assuming $\delta < 1/6$, we have with probability at least $2/3 - \delta \geq 1/2$, one of the iterates of SGD can achieve training loss within $\epsilon'$ of optimal. This completes the proof. $\square$

## A.5 Proof of Corollary 3.7

*Proof of Corollary 3.7.* Recall the condition in Theorem 3.6:

$$\frac{\sigma_{\min}^2(\mathbf{A})\sigma_{\min}^2(\mathbf{B})}{\|\mathbf{A}\|_2\|\mathbf{B}\|_2} \geq C \cdot \frac{n\|\mathbf{X}\|_2 \cdot \log(L(\mathbf{W}^{(0)})/\epsilon)}{B\sigma_r^2(\mathbf{X})} \cdot \sqrt{L(\mathbf{W}^{(0)})}, \tag{A.17}$$

Then plugging in the results in Proposition 3.3 and the fact that $\|\mathbf{X}\|_F \leq \sqrt{r}\|\mathbf{X}\|_2$, we obtain that condition (A.17) can be satisfied if $m = O\left(kr\kappa^2 \log^2(1/\epsilon) \cdot B/n\right)$.

In addition, consider sufficiently small $\epsilon$ such that $\epsilon \leq \tilde{O}\left(B\|\mathbf{X}\|_{2,\infty}^2/(n\|\mathbf{X}\|_2^2)\right)$, then and use the fact that $\|\mathbf{X}\|_{2,\infty} \leq \|\mathbf{X}\|_2$ we have $\eta = O\left(kB\epsilon/(Lmn\kappa\|\mathbf{X}\|_2^2)\right)$ based on the results in Proposition 3.3. Then in order to achieve $\epsilon$-suboptimal training loss, the iteration complexity is

$$T = \frac{e}{\eta L\sigma_{\min}^2 \sigma_{\min}^2(\mathbf{B})\sigma_r^2(\mathbf{X})} \log\left(\frac{L(\mathbf{W}^{(0)} - L(\mathbf{W}^*))}{\epsilon}\right) = O\left(\kappa^2\epsilon^{-1}\log(1/\epsilon) \cdot n/B\right).$$

This completes the proof. $\square$

## A.6 Proof of Theorem 3.8

*Proof of Theorem 3.8.* Similar to the proof of Theorem 3.6, we set the neural network width and step size as follows,

$$\frac{\mu_A^2 \mu_B^2}{\lambda_A \lambda_B} \geq \frac{4\sqrt{2e^3}n\|\mathbf{X}\|_2}{B\sigma_r^2(\mathbf{X})} \cdot \sqrt{2L(\mathbf{W}^{(0)})}$$

$$\eta \leq \frac{\log(2)B^2 \mu_A^2 \mu_B^2(\mathbf{B})\sigma_r^2(\mathbf{X})}{54e^3 Ln^2 \lambda_A^4 \lambda_B^4 \|\mathbf{X}\|_2^4 \cdot \log(T/\delta)},$$

where $\lambda_A$, $\mu_A$, $\lambda_B$ and $\mu_B$ denote $\|\mathbf{A}\|_2$, $\sigma_{\min}(\mathbf{A})$, $\|\mathbf{B}\|_2$ and $\sigma_{\min}(\mathbf{B})$ respectively.

Different from the proof of Theorem 3.6, the convergence guarantee established in this regime is made on the last iterate of SGD, rather than the best one. Besides, we will prove the theorem by induction on the update parameter $t$, using the following two-part inductive hypothesis:

(i) $\max_{l\in[L]}\|\mathbf{W}_l^{(t)}\|_F \leqslant 0.5/L$

(ii) $L(\mathbf{W}^{(t)}) \leqslant 2L(\mathbf{W}^{(0)}) \cdot \left(1 - \frac{s\eta L\mu_A^2\mu_B^2\sigma_r^2(\mathbf{X})}{e}\right)^s$.

**Induction for Part (i)** We first prove that $\max_{l\in[L]}\|\mathbf{W}_l^{(t)}\|_F \leqslant 0.5/L$. By triangle inequality and the update rule of SGD, we have

$$
\begin{aligned}
\|\mathbf{W}_l^{(t)}\|_F &\leqslant \sum_{s=0}^{t-1} \eta\|\mathbf{G}_l\|_F \\
&\leqslant \eta\sum_{s=0}^{t-1} \frac{\sqrt{2e}n\lambda_A\lambda_B\|\mathbf{X}\|_2}{B}\left(L(\mathbf{W}^{(s)}) - L(\mathbf{W}^*)\right)^{1/2} \\
&\leqslant \frac{\sqrt{2e}\eta n\lambda_A\lambda_B\|\mathbf{X}\|_2}{B} \cdot \left(L(\mathbf{W}^{(0)}) - L(\mathbf{W}^*)\right)^{1/2} \cdot \sum_{s=0}^{t-1}\left(1 - \frac{\eta L\mu_A^2\mu_B^2\sigma_r^2(\mathbf{X})}{2e}\right)^s \\
&\leqslant \frac{\sqrt{8e^3}n\lambda_A\lambda_B\|\mathbf{X}\|_2}{BL\mu_A^2\mu_B^2\sigma_r^2(\mathbf{X})} \cdot \left(L(\mathbf{W}^{(0)}) - L(\mathbf{W}^*)\right)^{1/2}
\end{aligned}
$$

where the second inequality is by Lemma A.1, the third inequality follows from Part (ii) for all $s < t$ and the fact that $(1-x)^{1/2} \leqslant 1 - x/2$ for all $x \in [0,1]$. Then applying our choice of $m$ implies that $\|\mathbf{W}_l^{(t)}\|_F \leqslant 0.5/L$.

**Induction for Part (ii)** Similar to Part (ii) and (iii) of the induction step in the proof of Theorem 3.6, we first prove the convergence in expectation, and then use Azuma's inequality to get the high-probability based results. It can be simply verfied that

$$
\lambda_A\lambda_B \geqslant \frac{\mu_A^2\mu_B^2}{\lambda_A\lambda_B} \geqslant \frac{4\sqrt{2e^3}n\|\mathbf{X}\|_2 \cdot \log(L(\mathbf{W}^{(0)})/\epsilon)}{B\sigma_r^2(\mathbf{X})} \cdot \sqrt{2L(\mathbf{W}^{(0)})} \geqslant \frac{2\sqrt{2e^{-1}L(\mathbf{W}^{(0)})}}{\|\mathbf{X}\|_2}
$$

$$
\eta \leqslant \frac{\log(2)B^2\mu_A^2\mu_B^2(\mathbf{B})\sigma_r^2(\mathbf{X})}{96e^3Ln^2\lambda_A^4\lambda_B^4\|\mathbf{X}\|_2^4 \cdot \log(T/\delta)} \leqslant \frac{B\mu_A^2\mu_B^2\sigma_r^2(\mathbf{X})}{6e^3Ln\lambda_A^4\lambda_B^4\|\mathbf{X}\|_2^2\|\mathbf{X}\|_{2,\infty}^2}.
$$

Thus, we can leverage (A.12) and obtain

$$
\mathbb{E}\left[L(\mathbf{W}^{(t)})|\mathbf{W}^{(t-1)}\right] - L(\mathbf{W}^{(t-1)}) \leqslant -\frac{4\eta L\mu_A^2\mu_B^2\sigma_r^2(\mathbf{X})}{3e}L(\mathbf{W}^{(t-1)}),
$$

where we use the fact that $L(\mathbf{W}^*) = 0$. Then by Jensen's inequality, we have

$$
\begin{aligned}
\mathbb{E}\left[\log\left(L(\mathbf{W}^{(t)})\right)|\mathbf{W}^{(t-1)}\right] &\leqslant \log\left(L(\mathbf{W}^{(t-1)})\right) + \log\left(1 - \frac{4\eta L\mu_A^2\mu_B^2\sigma_r^2(\mathbf{X})}{3e}\right), \\
&\leqslant \log\left(L(\mathbf{W}^{(t-1)})\right) - \frac{4\eta L\mu_A^2\mu_B^2\sigma_r^2(\mathbf{X})}{3e},
\end{aligned}
$$

where the second inequality is by $\log(1+x) \leqslant x$. Then similar to the proof of Theorem 3.6, we are going to apply martingale inequality to prove this part. Let $\mathcal{F}_t = \sigma\{\mathbf{W}^{(0)}, \cdots, \mathbf{W}^{(t)}\}$ be a $\sigma$-algebra, and $\mathbb{F} = \{\mathcal{F}_t\}_{t\geqslant 1}$ be a filtration, the above inequality implies that

$$
\mathbb{E}\left[\log\left(L(\mathbf{W}^{(t)})\right)|\mathcal{F}_{t-1}\right] + \frac{4t\eta L\mu_A^2\mu_B^2\sigma_r^2(\mathbf{X})}{3e} \leqslant \log\left(L(\mathbf{W}^{(t-1)})\right) + \frac{4(t-1)\eta L\mu_A^2\mu_B^2\sigma_r^2(\mathbf{X})}{3e}, \tag{A.18}
$$

which implies that $\left\{\log\left(L(\mathbf{W}^{(t)})\right) + 4t\eta L\mu_A^2\mu_B^2\sigma_r^2(\mathbf{X})/(3e)\right\}$ is a super-martingale. Besides, by (A.15), we can obtain

$$
\log\left(L(\mathbf{W}^{(t)})\right) \leqslant \log\left(L(\mathbf{W}^{(t-1)})\right) + \frac{3e\eta Ln\lambda_A^2\lambda_B^2\|\mathbf{X}\|_2^2}{B},
$$

which implies that

$$
\begin{aligned}
&\log\left(L(\mathbf{W}^{(t)})\right) + \frac{4t\eta L\mu_A^2\mu_B^2\sigma_r^2(\mathbf{X})}{3e} \\
&\leqslant \log\left(L(\mathbf{W}^{(t-1)})\right) + \frac{4(t-1)\eta L\mu_A^2\mu_B^2\sigma_r^2(\mathbf{X})}{3e} + \frac{4e\eta Ln\lambda_A^2\lambda_B^2\|\mathbf{X}\|_2^2}{B},
\end{aligned}
$$

where we again use the fact that $\log(1 + x) \leqslant x$. Thus, by the one-sided Azuma's inequality we have with probability at least $1 - \delta'$ that

$$
\begin{aligned}
\log\left(L(\mathbf{W}^{(t)})\right) &\leqslant \log\left(L(\mathbf{W}^{(0)})\right) - \frac{4t\eta L\mu_A^2\mu_B^2\sigma_r^2(\mathbf{X})}{3e} + \frac{4e\eta Ln\lambda_A^2\lambda_B^2\|\mathbf{X}\|_2^2}{B} \cdot \sqrt{2t\log(1/\delta')} \\
&\leqslant \log\left(L(\mathbf{W}^{(0)})\right) - \frac{t\eta L\mu_A^2\mu_B^2\sigma_r^2(\mathbf{X})}{e} + \frac{96e^3\eta Ln^2\lambda_A^4\lambda_B^4\|\mathbf{X}\|_2^4\log(1/\delta')}{B^2\mu_A^2\mu_B^2\sigma_r^2(\mathbf{X})} \\
&\leqslant \log\left(L(\mathbf{W}^{(0)})\right) - \frac{t\eta L\mu_A^2\mu_B^2\sigma_r^2(\mathbf{X})}{e} + \log(2),
\end{aligned}
$$

where the second inequality follows from the fact that $-at + b\sqrt{t} \leqslant b^2/a$, and the last inequality is by our choice of $\eta$ that

$$
\eta \leqslant \frac{\log(2)B^2\mu_A^2\mu_B^2(\mathbf{B})\sigma_r^2(\mathbf{X})}{96e^3Ln^2\lambda_A^4\lambda_B^4\|\mathbf{X}\|_2^4 \cdot \log(1/\delta')}.
$$

Then it is clear that with probability at least $1 - \delta'$,

$$
L(\mathbf{W}^{(t)}) \leqslant 2L(\mathbf{W}^{(0)}) \cdot \exp\left(- \frac{t\eta L\mu_A^2\mu_B^2\sigma_r^2(\mathbf{X})}{e}\right), \tag{A.19}
$$

which completes the induction for Part (ii).

Similar to the proof of Theorem 3.6, (A.19) holds with probability at least $1 - \delta'$ for a given $t$. Then we can set $\delta' = \delta/T$ and apply union bound such that with probability at least $1 - \delta$, (A.19) holds for all $t \leqslant T$. This completes the proof. □

## A.7 Proof of Corollary 3.9

*Proof of Corollary 3.9.* Recall the condition in Theorem 3.8:

$$
\frac{\sigma_{\min}^2(\mathbf{A})\sigma_{\min}^2(\mathbf{B})}{\|\mathbf{A}\|_2\|\mathbf{B}\|_2} \geqslant C \cdot \frac{n\|\mathbf{X}\|_2}{B\sigma_r^2(\mathbf{X})} \cdot \sqrt{L(\mathbf{W}^{(0)})}, \tag{A.20}
$$

Then plugging in the results in Proposition 3.3 and the fact that $\|\mathbf{X}\|_F \leqslant \sqrt{r}\|\mathbf{X}\|_2$, we obtain that condition (A.17) can be satisfied if $m = O\left(kr\kappa^2 \cdot B/n\right)$.

In addition, it can be computed that $\eta = O\left(kB^2/(Lmn^2\kappa\|\mathbf{X}\|_2^2)\right)$ based on the results in Proposition 3.3. Then in order to achieve $\epsilon$-suboptimal training loss, the iteration complexity is

$$
T = \frac{e}{\eta L\sigma_{\min}^2\sigma_{\min}^2(\mathbf{B})\sigma_r^2(\mathbf{X})}\log\left(\frac{L(\mathbf{W}^{(0)}) - L(\mathbf{W}^*)}{\epsilon}\right) = O\left(\kappa^2\log(1/\epsilon) \cdot n^2/B^2\right).
$$

This completes the proof. □

# B Proofs of Technical Lemmas

## B.1 Proof of Lemma A.1

We first note the following useful lemmas.

**Lemma B.1** (Claim B.1 in Du & Hu (2019))**.** Define $\mathbf{\Phi} = \arg\min_{\mathbf{\Theta}\in\mathbb{R}^{k\times d}}\|\mathbf{\Theta X} - \mathbf{Y}\|_F^2$, then for any $\mathbf{U} \in \mathbb{R}^{k\times d}$ it holds that

$$
\|\mathbf{UX} - \mathbf{Y}\|_F^2 = \|\mathbf{UX} - \mathbf{\Phi X}\|_F^2 + \|\mathbf{\Phi X} - \mathbf{Y}\|_F^2.
$$

**Lemma B.2** (Theorem 1 in Fang et al. (1994))**.** Let $\mathbf{U}, \mathbf{V} \in \mathbb{R}^{d\times d}$ be two positive definite matrices, then it holds that

$$
\lambda_{\min}(\mathbf{U})\text{Tr}(\mathbf{V}) \leqslant \text{Tr}(\mathbf{UV}) \leqslant \lambda_{\max}(\mathbf{U})\text{Tr}(\mathbf{V}).
$$

The following lemma is proved in Section B.3.

**Lemma B.3.** Let $\mathbf{U} \in \mathbb{R}^{d \times r}$ be a rank-$r$ matrix. Then for any $\mathbf{V} \in \mathbb{R}^{r \times k}$, it holds that

$$\sigma_{\min}(\mathbf{U})\|\mathbf{V}\|_F \leqslant \|\mathbf{U}\mathbf{V}\|_F \leqslant \sigma_{\max}(\mathbf{U})\|\mathbf{V}\|_F.$$

*Proof of Lemma A.1.* **Proof of gradient lower bound:** We first prove the gradient lower bound. Let $\mathbf{U} = \mathbf{B}(\mathbf{I} + \mathbf{W}_L) \dots (\mathbf{I} + \mathbf{W}_1)\mathbf{A}$, by Lemma B.1 and the definition of $L(\mathbf{W}^*)$, we know that there exist a matrix $\boldsymbol{\Phi} \in \mathbb{R}^{k \times d}$ such that

$$L(\mathbf{W}) = \frac{1}{2}\|\mathbf{U}\mathbf{X} - \boldsymbol{\Phi}\mathbf{X}\|_F^2 + L(\mathbf{W}^*). \tag{B.1}$$

Therefore, based on the assumption that $\max_{l \in [L]} \|\mathbf{W}_l\|_F \leqslant 0.5/L$, we have

$$\begin{aligned}
\|\nabla_{\mathbf{W}_l} L(\mathbf{W})\|_F^2 &= \left\|\left[\mathbf{B}(\mathbf{I} + \mathbf{W}_L)\cdots(\mathbf{I} + \mathbf{W}_{l+1})\right]^\top (\mathbf{U}\mathbf{X} - \boldsymbol{\Phi}\mathbf{X})\left[(\mathbf{I} + \mathbf{W}_{l-1})\cdots\mathbf{A}\mathbf{X}\right]^\top\right\|_F^2 \\
&\geqslant \sigma_{\min}^2((\mathbf{I} + \mathbf{W}_L)\cdots(\mathbf{I} + \mathbf{W}_{l+1})) \cdot \sigma_{\min}^2((\mathbf{I} + \mathbf{W}_{l-1})\cdots(\mathbf{I} + \mathbf{W}_1)) \\
&\quad \cdot \|\mathbf{B}^\top(\mathbf{U} - \boldsymbol{\Phi})\mathbf{X}\mathbf{X}^\top\mathbf{A}^\top\|_F^2 \\
&\geqslant (1 - 0.5/L)^{2L-2}\|\mathbf{B}^\top(\mathbf{U} - \boldsymbol{\Phi})\mathbf{X}\mathbf{X}^\top\mathbf{A}^\top\|_F^2,
\end{aligned}$$

where the last inequality follows from the fact that $\sigma_{\min}(\mathbf{I} + \mathbf{W}_l) \geqslant 1 - \|\mathbf{W}_l\|_2 \geqslant 1 - \|\mathbf{W}_l\|_F \geqslant 1 - 0.5/L$. Applying Lemma B.2, we get

$$\begin{aligned}
\|\mathbf{B}^\top(\mathbf{U} - \boldsymbol{\Phi})\mathbf{X}\mathbf{X}^\top\mathbf{A}^\top\|_F^2 &= \text{Tr}\big(\mathbf{B}\mathbf{B}^\top(\mathbf{U} - \boldsymbol{\Phi})\mathbf{X}\mathbf{X}^\top\mathbf{A}^\top\mathbf{A}\mathbf{X}\mathbf{X}^\top(\mathbf{U} - \boldsymbol{\Phi})^\top\big) \\
&\geqslant \lambda_{\min}(\mathbf{B}\mathbf{B}^\top) \cdot \text{Tr}\big(\mathbf{A}^\top\mathbf{A}\mathbf{X}\mathbf{X}^\top(\mathbf{U} - \boldsymbol{\Phi})^\top(\mathbf{U} - \boldsymbol{\Phi})\mathbf{X}\mathbf{X}^\top\big) \\
&\geqslant \lambda_{\min}(\mathbf{B}\mathbf{B}^\top) \cdot \lambda_{\min}(\mathbf{A}^\top\mathbf{A}) \cdot \|(\mathbf{U} - \boldsymbol{\Phi})\mathbf{X}\mathbf{X}^\top\|_F^2.
\end{aligned}$$

Note that $\mathbf{X}$ is of $r$-rank, thus there exists a full-rank matrix $\widetilde{\mathbf{X}} \in \mathbb{R}^{d \times r}$ such that $\widetilde{\mathbf{X}}\widetilde{\mathbf{X}}^\top = \mathbf{X}\mathbf{X}^\top$. Thus we have

$$\|(\mathbf{U} - \boldsymbol{\Phi})\mathbf{X}\|_F^2 = \text{Tr}\big((\mathbf{U} - \boldsymbol{\Phi})\mathbf{X}\mathbf{X}^\top(\mathbf{U} - \phi)^\top\big) = \text{Tr}\big((\mathbf{U} - \boldsymbol{\Phi})\widetilde{\mathbf{X}}\widetilde{\mathbf{X}}^\top(\mathbf{U} - \phi)^\top\big) = \big\|(\mathbf{U} - \boldsymbol{\Phi})\widetilde{\mathbf{X}}\big\|_F^2. \tag{B.2}$$

Therefore,

$$\begin{aligned}
\|(\mathbf{U} - \boldsymbol{\Phi})\mathbf{X}\mathbf{X}^\top\|_F^2 &= \big\|(\mathbf{U} - \boldsymbol{\Phi})\widetilde{\mathbf{X}}\widetilde{\mathbf{X}}^\top\big\|_F^2 \\
&= \text{Tr}\big((\mathbf{U} - \boldsymbol{\Phi})\widetilde{\mathbf{X}}\widetilde{\mathbf{X}}^\top\widetilde{\mathbf{X}}\widetilde{\mathbf{X}}^\top(\mathbf{U} - \boldsymbol{\Phi})^\top\big) \\
&\geqslant \lambda_{\min}(\widetilde{\mathbf{X}}^\top\widetilde{\mathbf{X}}) \cdot \|(\mathbf{U} - \boldsymbol{\Phi})\widetilde{\mathbf{X}}\|_F^2 \\
&= 2\sigma_r^2(\mathbf{X}) \cdot (L(\mathbf{W}) - L(\mathbf{W}^*)), \tag{B.3}
\end{aligned}$$

where the inequality follows from Lemma B.2 and the last equality follows from (B.2), (B.1) and the fact that $\lambda_{\min}(\widetilde{\mathbf{X}}^\top\widetilde{\mathbf{X}}) = \lambda_r(\mathbf{X}\mathbf{X}^\top) = \sigma_r^2(\mathbf{X})$. Note that we assume $d, k \leqslant m$ and $d \leqslant n$. Thus it follows that $\lambda_{\min}(\mathbf{B}\mathbf{B}^\top) = \sigma_{\min}^2(\mathbf{B})$ and $\lambda_{\min}(\mathbf{A}^\top\mathbf{A}) = \sigma_{\min}^2(\mathbf{A})$. Then putting everything together, we can obtain

$$\|\nabla_{\mathbf{W}_l} L(\mathbf{W})\|_F^2 \geqslant 2\sigma_{\min}^2(\mathbf{B})\sigma_{\min}^2(\mathbf{A})\sigma_r^2(\mathbf{X})(1 - 0.5/L)^{2L-2}\big(L(\mathbf{W} - L(\mathbf{W}^*)\big).$$

Then using the inequality $(1 - 0.5/L)^{2L-2} \geqslant e^{-1}$, we are able to complete the proof of gradient lower bound.

**Proof of gradient upper bound:** The gradient upper bound can be proved in a similar way. Specifically, Lemma B.3 implies

$$\begin{aligned}
\|\nabla_{\mathbf{W}_l} L(\mathbf{W})\|_F^2 &= \left\|\left[\mathbf{B}(\mathbf{I} + \mathbf{W}_L)\cdots(\mathbf{I} + \mathbf{W}_{l+1})\right]^\top (\mathbf{U}\mathbf{X} - \boldsymbol{\Phi}\mathbf{X})\left[(\mathbf{I} + \mathbf{W}_{l-1})\cdots\mathbf{A}\mathbf{X}\right]^\top\right\|_F^2 \\
&\leqslant \sigma_{\max}^2((\mathbf{I} + \mathbf{W}_L)\cdots(\mathbf{I} + \mathbf{W}_{l+1})) \cdot \sigma_{\max}^2((\mathbf{I} + \mathbf{W}_{l-1})\cdots(\mathbf{I} + \mathbf{W}_1)) \\
&\quad \cdot \|\mathbf{B}^\top(\mathbf{U} - \boldsymbol{\Phi})\mathbf{X}\mathbf{X}^\top\mathbf{A}^\top\|_F^2 \\
&\leqslant \sigma_{\max}^2((\mathbf{I} + \mathbf{W}_L)\cdots(\mathbf{I} + \mathbf{W}_{l+1})) \cdot \sigma_{\max}^2((\mathbf{I} + \mathbf{W}_{l-1})\cdots(\mathbf{I} + \mathbf{W}_1)) \\
&\quad \cdot \|\mathbf{B}\|_2^2\|\mathbf{A}\|_2^2 \cdot \|(\mathbf{U} - \boldsymbol{\Phi})\mathbf{X}\mathbf{X}^\top\|_F^2 \\
&\leqslant (1 + 0.5/L)^{2L-2}\|\mathbf{B}\|_2^2\|\mathbf{A}\|_2^2 \cdot \|(\mathbf{U} - \boldsymbol{\Phi})\mathbf{X}\mathbf{X}^\top\|_F^2,
\end{aligned}$$

where the last inequality is by the assumption that $\max_{l\in[L]}\|\mathbf{W}_l\|_F \leqslant 0.5/L$. By (B.3), we have

$$
\begin{aligned}
\|(\mathbf{U}-\mathbf{\Phi})\mathbf{X}\mathbf{X}^\top\|_F^2 &= \left\|(\mathbf{U}-\mathbf{\Phi})(\mathbf{X}\mathbf{X}^\top)^{1/2}(\mathbf{X}\mathbf{X}^\top)^{1/2}\right\|_F^2 \\
&\leqslant \lambda_{\max}(\mathbf{X}\mathbf{X}^\top)\cdot\|(\mathbf{U}-\mathbf{\Phi})(\mathbf{X}\mathbf{X}^\top)^{1/2}\|_F^2 \\
&= \lambda_{\max}(\mathbf{X}\mathbf{X}^\top)\cdot\|(\mathbf{U}-\mathbf{\Phi})\mathbf{X}\|_F^2 \\
&= 2\|\mathbf{X}\|_2^2\cdot(L(\mathbf{W})-L(\mathbf{W}^*)),
\end{aligned}
$$

where the inequality is by Lemma B.3 and the second equality is by (B.2). Therefore, combining the above results yields

$$
\|\nabla_{\mathbf{W}_l}L(\mathbf{W})\|_F^2 \leqslant 2\sigma_{\max}^2(\mathbf{B})\sigma_{\max}^2(\mathbf{A})\|\mathbf{X}\|_2^2(1+0.5/L)^{2L-2}\big(L(\mathbf{W}-L(\mathbf{W}^*)\big).
$$

Using the inequality $(1+0.5/L)^{2L-2} \leqslant (1+0.5/L)^{2L} \leqslant e$, we are able to complete the proof of gradient upper bound.

**Proof of the upper bound of $\|\nabla_{\mathbf{W}_l}\ell(\mathbf{W};\mathbf{x}_i,\mathbf{y}_i)\|_F^2$:** Let $\mathbf{U} = \mathbf{B}(\mathbf{I}+\mathbf{W}_L)\cdots(\mathbf{I}+\mathbf{W}_1)\mathbf{A}$, we have

$$
\nabla_{\mathbf{W}_l}\ell(\mathbf{W};\mathbf{x}_i,\mathbf{y}_i) = \big[\mathbf{B}(\mathbf{I}+\mathbf{W}_L)\cdots(\mathbf{I}+\mathbf{W}_{l+1})\big]^\top(\mathbf{U}\mathbf{x}_i-\mathbf{y}_i)\big[(\mathbf{I}+\mathbf{W}_{l-1})\cdots\mathbf{A}\bar{\mathbf{x}}_i\big]^\top.
$$

Therefore, by Lemma B.3, we have

$$
\begin{aligned}
\|\nabla_{\mathbf{W}_l}\ell(\mathbf{W};\mathbf{x}_i,\mathbf{y}_i)\|_F^2 &\leqslant \sigma_{\max}^2\big((\mathbf{I}+\mathbf{W}_L)\cdot(\mathbf{I}+\mathbf{W}_{l+1})\big)\cdot\sigma_{\max}^2\big((\mathbf{I}+\mathbf{W}_{l-1})\cdots(\mathbf{I}+\mathbf{W}_1)\big) \\
&\quad\cdot\|\mathbf{B}^\top(\mathbf{U}\mathbf{x}_i-\mathbf{y}_i)\mathbf{x}_i\mathbf{A}^\top\|_F^2 \\
&\leqslant (1+0.5/L)^{2L-2}\cdot\|\mathbf{B}\|_2^2\|\mathbf{A}\|_2^2\|\mathbf{x}_i\|_2^2\cdot\|\mathbf{U}\mathbf{x}_i-\mathbf{y}_i\|_F^2 \\
&\leqslant 2e\|\mathbf{A}\|_2^2\|\mathbf{B}\|_2^2\|\mathbf{x}_i\|_2^2\ell(\mathbf{W};\mathbf{x}_i,\mathbf{y}_i),
\end{aligned}
$$

where the last inequality is by the fact that $(1+0.5/L)^{2L-2} \leqslant e$.

**Proof of the upper bound of stochastic gradient:** Define by $\mathcal{B}$ the set of training data points used to compute the stochastic gradient, then define by $\bar{\mathbf{X}}$ and $\bar{\mathbf{Y}}$ the stacking of $\{\mathbf{x}_i\}_{i\in\mathcal{B}}$ and $\{\mathbf{y}_i\}_{i\in\mathcal{B}}$ respectively. Let $\mathbf{U} = \mathbf{B}(\mathbf{I}+\mathbf{W}_L)\cdots(\mathbf{I}+\mathbf{W}_1)\mathbf{A}$, the minibatch stochastic gradient takes form

$$
\begin{aligned}
\mathbf{G}_l &= \frac{n}{B}\sum_{i\in\mathcal{B}}\nabla_{\mathbf{W}_l}\ell(\mathbf{W};\mathbf{x}_i,\mathbf{y}_i) \\
&= \frac{n}{B}\big[\mathbf{B}(\mathbf{I}+\mathbf{W}_L)\cdots(\mathbf{I}+\mathbf{W}_{l+1})\big]^\top(\mathbf{U}\bar{\mathbf{X}}-\bar{\mathbf{Y}})\big[(\mathbf{I}+\mathbf{W}_{l-1})\cdots\mathbf{A}\bar{\mathbf{X}}\big]^\top.
\end{aligned}
$$

Then by Lemma B.3, we have

$$
\begin{aligned}
\|\mathbf{G}_l\|_F^2 &\leqslant \frac{n^2}{B^2}\sigma_{\max}^2\big((\mathbf{I}+\mathbf{W}_L)\cdot(\mathbf{I}+\mathbf{W}_{l+1})\big)\cdot\sigma_{\max}^2\big((\mathbf{I}+\mathbf{W}_{l-1})\cdots(\mathbf{I}+\mathbf{W}_1)\big) \\
&\quad\cdot\|\mathbf{B}^\top(\mathbf{U}\bar{\mathbf{X}}-\bar{\mathbf{Y}})\bar{\mathbf{X}}^\top\mathbf{A}^\top\|_F^2 \\
&\leqslant \frac{n^2}{B^2}\cdot(1+0.5/L)^{2L-2}\cdot\|\mathbf{B}\|_2^2\|\mathbf{A}\|_2^2\|\bar{\mathbf{X}}\|_2^2\cdot\|\mathbf{U}\bar{\mathbf{X}}-\bar{\mathbf{Y}}\|_F^2 \\
&\leqslant \frac{en^2}{B^2}\|\mathbf{B}\|_2^2\|\mathbf{A}\|_2^2\|\bar{\mathbf{X}}\|_2^2\cdot\|\mathbf{U}\bar{\mathbf{X}}-\bar{\mathbf{Y}}\|_F^2.
\end{aligned}
$$

where the second inequality is by the assumptions that $\max_{l\in[L]}\|\mathbf{W}_l\|_F \leqslant 0.5/L$, and the last inequality follows from the the fact that $(1+0.5/L)^{2L-2} \leqslant (1+0.5/L)^{2L} \leqslant e$. Note that $\bar{\mathbf{X}}$ and $\bar{\mathbf{Y}}$ are constructed by stacking $B$ columns from $\mathbf{X}$ and $\mathbf{Y}$ respectively, thus we have $\|\bar{\mathbf{X}}\|_2^2 \leqslant \|\mathbf{X}\|_2^2$ and $\|\mathbf{U}\bar{\mathbf{X}}-\bar{\mathbf{Y}}\|_F^2 \leqslant \|\mathbf{U}\mathbf{X}-\mathbf{Y}\|_F^2 = 2L(\mathbf{W})$. Then it follows that

$$
\|\mathbf{G}_l\|_F^2 \leqslant \frac{2en^2}{B^2}\|\mathbf{B}\|_2^2\|\mathbf{A}\|_2^2\|\mathbf{X}\|_2^2\cdot L(\mathbf{W}).
$$

This completes the proof of the upper bound of stochastic gradient. $\qquad\square$

## B.2 PROOF OF LEMMA A.2

*Proof of Lemma A.2.* Let $\mathbf{U} = \mathbf{B}(\mathbf{I} + \mathbf{W}_L) \cdots (\mathbf{I} + \mathbf{W}_1)\mathbf{A}$ and $\widetilde{\mathbf{U}} = \mathbf{B}(\mathbf{I} + \widetilde{\mathbf{W}}_L) \cdots (\mathbf{I} + \widetilde{\mathbf{W}}_1)\mathbf{A}$ and $\mathbf{\Delta} = \widetilde{\mathbf{U}} - \mathbf{U}$. We have

$$
\begin{aligned}
L(\widetilde{\mathbf{W}}) - L(\mathbf{W}) &= \frac{1}{2}\big(\|\widetilde{\mathbf{U}}\mathbf{X} - \mathbf{Y}\|_F^2 - \|\mathbf{U}\mathbf{X} - \mathbf{Y}\|_F^2\big) \\
&= \frac{1}{2}\big(\|(\mathbf{U} + \mathbf{\Delta})\mathbf{X} - \mathbf{Y}\|_F^2 - \|\mathbf{U}\mathbf{X} - \mathbf{Y}\|_F^2\big) \\
&= \frac{1}{2}\big(\|\mathbf{U}\mathbf{X} - \mathbf{Y} + \mathbf{\Delta}\mathbf{X}\|_F^2 - \|\mathbf{U}\mathbf{X} - \mathbf{Y}\|_F^2\big) \\
&= \frac{1}{2}\big(\|2\langle \mathbf{U}\mathbf{X} - \mathbf{Y}, \mathbf{\Delta}\mathbf{X}\rangle + \|\mathbf{\Delta}\mathbf{X}\|_F^2\big) \\
&= \langle \mathbf{U}\mathbf{X} - \mathbf{Y}, (\widetilde{\mathbf{U}} - \mathbf{U})\mathbf{X}\rangle + \frac{1}{2}\|(\widetilde{\mathbf{U}} - \mathbf{U})\mathbf{X}\|_F^2.
\end{aligned}
\tag{B.4}
$$

We begin by working on the first term. Let $\mathbf{V} = (\mathbf{I} + \mathbf{W}_L) \cdots (\mathbf{I} + \mathbf{W}_1)$ and $\widetilde{\mathbf{V}} = (\mathbf{I} + \widetilde{\mathbf{W}}_L) \cdots (\mathbf{I} + \widetilde{\mathbf{W}}_1)$, so that $\widetilde{\mathbf{U}} - \mathbf{U} = \mathbf{B}(\widetilde{\mathbf{V}} - \mathbf{V})\mathbf{A}$. Breaking down the effect of transforming $\mathbf{V} = \prod_{j=L}^{1}(\mathbf{I} + \mathbf{W}_j)$ into $\widetilde{\mathbf{V}} = \prod_{j=L}^{1}(\mathbf{I} + \widetilde{\mathbf{W}}_j)$ into the effects of replacing one layer at a time, we get

$$
\widetilde{\mathbf{V}} - \mathbf{V} = \sum_{l=1}^{L}\left[\left(\prod_{j=L}^{l+1}(\mathbf{I} + \mathbf{W}_j)\right)\left(\prod_{j=l}^{1}(\mathbf{I} + \widetilde{\mathbf{W}}_j)\right) - \left(\prod_{j=L}^{l}(\mathbf{I} + \mathbf{W}_j)\right)\left(\prod_{j=l-1}^{1}(\mathbf{I} + \widetilde{\mathbf{W}}_j)\right)\right]
$$

and, for each $l$, pulling out a common factor of $\left(\prod_{j=L}^{l+1}(\mathbf{I} + \mathbf{W}_j)\right)\left(\prod_{j=l}^{1-l}(\mathbf{I} + \widetilde{\mathbf{W}}_j)\right)$ gives

$$
\begin{aligned}
\widetilde{\mathbf{V}} - \mathbf{V} &= \sum_{l=1}^{L}(\mathbf{I} + \mathbf{W}_L) \cdots (\mathbf{I} + \mathbf{W}_{l+1})(\widetilde{\mathbf{W}}_l - \mathbf{W}_l)(\mathbf{I} + \widetilde{\mathbf{W}}_{l-1}) \cdots (\mathbf{I} + \widetilde{\mathbf{W}}_1) \\
&= \underbrace{\sum_{l=1}^{L}(\mathbf{I} + \mathbf{W}_L) \cdots (\mathbf{I} + \mathbf{W}_{l+1})(\widetilde{\mathbf{W}}_l - \mathbf{W}_l)(\mathbf{I} + \mathbf{W}_{l-1}) \cdots (\mathbf{I} + \mathbf{W}_1)}_{\mathbf{V}_1} \\
&\quad + \underbrace{\sum_{l=1}^{L}(\mathbf{I} + \mathbf{W}_L) \cdots (\mathbf{I} + \mathbf{W}_{l+1})(\widetilde{\mathbf{W}}_l - \mathbf{W}_l)}_{\mathbf{V}_2} \\
&\quad \cdot \underbrace{\big[(\mathbf{I} + \widetilde{\mathbf{W}}_{l-1}) \cdots (\mathbf{I} + \widetilde{\mathbf{W}}_1) - (\mathbf{I} + \mathbf{W}_{l-1}) \cdots (\mathbf{I} + \mathbf{W}_1)\big]}_{\mathbf{V}_2}.
\end{aligned}
\tag{B.5}
$$

The first term $\mathbf{V}_1$ satisfies

$$\langle \mathbf{UX} - \mathbf{Y}, \mathbf{BV}_1\mathbf{AX} \rangle$$

$$= \left\langle \mathbf{UX} - \mathbf{Y}, \mathbf{B}\left(\sum_{l=1}^{L}(\mathbf{I}+\mathbf{W}_L)\cdots(\mathbf{I}+\mathbf{W}_{l+1})(\widetilde{\mathbf{W}}_l - \mathbf{W}_l)(\mathbf{I}+\mathbf{W}_{l-1})\cdots(\mathbf{I}+\mathbf{W}_1)\right)\mathbf{AX} \right\rangle$$

$$= \sum_{l=1}^{L}\left\langle \mathbf{UX} - \mathbf{Y}, \mathbf{B}(\mathbf{I}+\mathbf{W}_L)\cdots(\mathbf{I}+\mathbf{W}_{l+1})(\widetilde{\mathbf{W}}_l - \mathbf{W}_l)(\mathbf{I}+\mathbf{W}_{l-1})\cdots(\mathbf{I}+\mathbf{W}_1)\mathbf{AX} \right\rangle$$

$$= \sum_{l=1}^{L}\mathrm{Tr}((\mathbf{UX} - \mathbf{Y})^{\top}\mathbf{B}(\mathbf{I}+\mathbf{W}_L)\cdots(\mathbf{I}+\mathbf{W}_{l+1})(\widetilde{\mathbf{W}}_l - \mathbf{W}_l)(\mathbf{I}+\mathbf{W}_{l-1})\cdots(\mathbf{I}+\mathbf{W}_1)\mathbf{AX})$$

$$= \sum_{l=1}^{L}\mathrm{Tr}((\mathbf{I}+\mathbf{W}_{l-1})\cdots(\mathbf{I}+\mathbf{W}_1)\mathbf{AX}(\mathbf{UX} - \mathbf{Y})^{\top}\mathbf{B}(\mathbf{I}+\mathbf{W}_L)\cdots(\mathbf{I}+\mathbf{W}_{l+1})(\widetilde{\mathbf{W}}_l - \mathbf{W}_l))$$

$$= \sum_{l=1}^{L}\left\langle [\mathbf{B}(\mathbf{I}+\mathbf{W}_L)\cdots(\mathbf{I}+\mathbf{W}_{l+1})]^{\top}(\mathbf{UX} - \mathbf{Y})[(\mathbf{I}+\mathbf{W}_{l-1})\cdots\mathbf{AX}]^{\top}, \widetilde{\mathbf{W}}_l - \mathbf{W}_l \right\rangle$$

$$= \sum_{l=1}^{L}\langle\nabla_{\mathbf{W}_l}L(\mathbf{W}), \widetilde{\mathbf{W}}_l - \mathbf{W}_l\rangle, \tag{B.6}$$

where the first equality is by the definition of $\mathbf{V}_1$. Now we focus on the second term $\mathbf{V}_2$ of (B.5),

$$\mathbf{V}_2 = \sum_{l=1}^{L}(\mathbf{I}+\mathbf{W}_L)\cdots(\mathbf{I}+\mathbf{W}_{l+1})(\widetilde{\mathbf{W}}_l - \mathbf{W}_l)$$

$$\cdot \sum_{s=1}^{l-1}(\mathbf{I}+\mathbf{W}_{l-1})\cdots(\mathbf{I}+\mathbf{W}_{s+1})(\widetilde{\mathbf{W}}_s - \mathbf{W}_s)(\mathbf{I}+\widetilde{\mathbf{W}}_{s-1})\cdots(\mathbf{I}+\widetilde{\mathbf{W}}_1).$$

Recalling that $\|\mathbf{W}_l\|_F, \|\widetilde{\mathbf{W}}_l\|_F \leqslant 0.5/L$ for all $l \in [L]$, by triangle inequality we have

$$\|\mathbf{V}_2\|_F \leqslant (1+0.5/L)^L \cdot \sum_{l,s\in[L]\,:\,l>s}\|\widetilde{\mathbf{W}}_l - \mathbf{W}_l\|_F \cdot \|\widetilde{\mathbf{W}}_s - \mathbf{W}_s\|_F$$

$$\leqslant (1+0.5/L)^L \cdot \left(\sum_{l=1}^{L}\|\widetilde{\mathbf{W}}_l - \mathbf{W}_l\|_F\right)^2,$$

where we use the fact that $\sum_{l,s\in[L]\,:\,l>s} a_l a_s \leqslant \sum_{l,s\in[L]} a_l a_s = \left(\sum_l a_l\right)^2$ holds for all $a_1, \ldots, a_L \geqslant 0$. Therefore, the following holds regarding $\mathbf{V}_2$:

$$\langle \mathbf{UX} - \mathbf{Y}, \mathbf{BV}_2\mathbf{AX} \rangle \leqslant \|\mathbf{UX} - \mathbf{Y}\|_F\|\mathbf{BV}_2\mathbf{AX}\|_F$$

$$\leqslant \sqrt{2L(\mathbf{W})}\|\mathbf{B}\|_2\|\mathbf{A}\|_2\|\mathbf{X}\|_2\|\mathbf{V}_2\|_F$$

$$\leqslant \sqrt{2e}\sqrt{L(\mathbf{W})}\|\mathbf{B}\|_2\|\mathbf{A}\|_2\|\mathbf{X}\|_2\left(\sum_{l=1}^{L}\|\widetilde{\mathbf{W}}_l - \mathbf{W}_l\|_F\right)^2 \tag{B.7}$$

where the third inequality follows from the fact that $(1+0.5/L)^L = (1+0.5/L)^L \leqslant \sqrt{e}$. Next, we are going to upper bound the second term of (B.4): $\frac{1}{2}\|(\widetilde{\mathbf{U}} - \mathbf{U})\mathbf{X}\|_F^2$. Note that, since $\|(\widetilde{\mathbf{U}} - \mathbf{U})\mathbf{X}\|_F^2 = \|\mathbf{B}(\widetilde{\mathbf{V}} - \mathbf{V})\mathbf{AX}\|_F^2 \leqslant \|\mathbf{A}\|_2^2\|\mathbf{B}\|_2^2\|\mathbf{X}\|_2^2\|\widetilde{\mathbf{V}} - \mathbf{V}\|_F^2$, it suffices to bound the norm $\|\widetilde{\mathbf{V}} - \mathbf{V}\|_F$. By (B.5), we have

$$\|\widetilde{\mathbf{V}} - \mathbf{V}\|_F = \left\|\sum_{l=1}^{L}(\mathbf{I}+\mathbf{W}_L)\cdots(\mathbf{I}+\mathbf{W}_{l+1})(\widetilde{\mathbf{W}}_l - \mathbf{W}_l)(\mathbf{I}+\widetilde{\mathbf{W}}_{l-1})\cdots(\mathbf{I}+\widetilde{\mathbf{W}}_1)\right\|_F$$

$$\leqslant (1+0.5/L)^L \sum_{l=1}^{L}\|\widetilde{\mathbf{W}}_l - \mathbf{W}_l\|_F. \tag{B.8}$$

Plugging (B.6), (B.7) and (B.8) into (B.4), we have

$$
\begin{aligned}
&L(\widetilde{\mathbf{W}}) - L(\mathbf{W}) \\
&= \big\langle \mathbf{U}\mathbf{X} - \mathbf{Y}, \mathbf{B}(\mathbf{V}_1 + \mathbf{V}_2)\mathbf{X} \big\rangle + \frac{1}{2}\big\|\mathbf{B}\big(\widetilde{\mathbf{V}} - \mathbf{V}\big)\mathbf{A}\mathbf{X}\big\|_F^2 \\
&\leqslant \sum_{l=1}^{L} \langle \nabla_{\mathbf{W}_l} L(\mathbf{W}), \widetilde{\mathbf{W}}_l - \mathbf{W}_l \rangle \\
&\qquad + \|\mathbf{A}\|_2 \|\mathbf{B}\|_2 \|\mathbf{X}\|_2 \big(\sqrt{2eL(\mathbf{W})} + 0.5e\|\mathbf{A}\|_2\|\mathbf{B}\|_2\|\mathbf{X}\|_2\big) \left(\sum_{l=1}^{L} \|\widetilde{\mathbf{W}}_l - \mathbf{W}_l\|_F\right)^2 \\
&\leqslant \sum_{l=1}^{L} \langle \nabla_{\mathbf{W}_l} L(\mathbf{W}), \widetilde{\mathbf{W}}_l - \mathbf{W}_l \rangle \\
&\qquad + L\|\mathbf{A}\|_2 \|\mathbf{B}\|_2 \|\mathbf{X}\|_2 \big(\sqrt{2eL(\mathbf{W})} + 0.5e\|\mathbf{A}\|_2\|\mathbf{B}\|_2\|\mathbf{X}\|_2\big) \sum_{l=1}^{L} \|\widetilde{\mathbf{W}}_l - \mathbf{W}_l\|_F^2, \qquad (\text{B.9})
\end{aligned}
$$

where the last inequality is by Jesen's inequality. This completes the proof.

$\square$

### B.3  PROOF OF LEMMA B.3

*Proof of Lemma B.3.*  Note that we have

$$
\|\mathbf{U}\mathbf{V}\|_F^2 = \text{Tr}(\mathbf{U}\mathbf{V}\mathbf{V}^\top\mathbf{U}^\top) = \text{Tr}(\mathbf{U}^\top\mathbf{U}\mathbf{V}\mathbf{V}^\top).
$$

By Lemma B.2, it is clear that

$$
\lambda_{\min}(\mathbf{U}^\top\mathbf{U})\text{Tr}(\mathbf{V}\mathbf{V}^\top) \leqslant \text{Tr}(\mathbf{U}^\top\mathbf{U}\mathbf{V}\mathbf{V}^\top) \leqslant \lambda_{\max}(\mathbf{U}^\top\mathbf{U})\text{Tr}(\mathbf{V}\mathbf{V}^\top).
$$

Since $\mathbf{U} \in \mathbb{R}^{d \times r}$ is of $r$-rank, thus we have $\lambda_{\min}(\mathbf{U}^\top\mathbf{U}) = \sigma_{\min}^2(\mathbf{U})$. Then applying the facts that $\lambda_{\max}(\mathbf{U}^\top\mathbf{U}) = \sigma_{\max}^2(\mathbf{U})$ and $\text{Tr}(\mathbf{V}\mathbf{V}^\top) = \|\mathbf{V}\|_F^2$, we are able to complete the proof.

$\square$

