# OpenReview forum: "On the Global Convergence  of Training Deep Linear ResNets"
_ICLR.cc/2020/Conference — Accept (Poster)_

### Official Review · AnonReviewer3 · 2019-10-23
**Official Blind Review #3**

**Rating:** 6

**Review:**

In this paper, the authors study the convergence of (stochastic) gradient descent in training deep linear residual networks, where linear transformation at input and output layers are fixed and matrices in other layers are trained. They first establish a global convergence of GD/SGD under some conditions on the fixed linear transformations. They they showed that for Gaussian random input and output transformation, global convergence still holds under conditions on the width of networks strictly milder than the literature. Linear convergence rate of SG/SGD are also established.

The paper is well written. The results seem novel and interesting.

It would be nice if the authors can give intuition why the input and output layer transformations need to be fixed in the analysis. What if happen if these matrices vary along the optimization process?

----------------------
After rebuttal:

I have read the authors' response. I would like to keep my original score.

**Experience Assessment:**

I have read many papers in this area.

**Review Assessment: Checking Correctness Of Derivations And Theory:**

I did not assess the derivations or theory.

**Review Assessment: Checking Correctness Of Experiments:**

I did not assess the experiments.

**Review Assessment: Thoroughness In Paper Reading:**

I read the paper at least twice and used my best judgement in assessing the paper.

---

> ### Author Response · Authors · 2019-11-11
> **Response to Review #3**
>
> Thanks for your supportive comments.
>
> Q1: “It would be nice if the authors can give intuition why the input and output layer transformations need to be fixed in the analysis. What if happen if these matrices vary along the optimization process?”
> A1: We believe that our analysis can be modified to address the case that A and B are trained (i.e., vary along the optimization process).  The high-level idea to prove this is: with proper initialization of A and B (e.g., Gaussian initialization with proper variance), we can show that they will stay close to their initial values similarly to the way that other layers were treated, and, given that they are close to their initial values, their singular values can be bounded above, and bounded away from zero, so that the objective function stays smooth, and they continue to propagate gradients effectively to the other layers.

---

### Official Review · AnonReviewer1 · 2019-10-23
**Official Blind Review #1**

**Rating:** 6

**Review:**

This paper studies the convergence properties of GD and SGD on deep linear resnets. The authors prove that, under certain conditions on the input and output transformations and with zero initialization, GD and SGD converges to global minima. The results derived in this paper show that the condition on the width of a deep linear resnet is less strict (by a factor O(L kappa) where L is the depth and kappa the condition number of the training data) that a network without residual connection. Overall, the paper is well-written and the analysis is fairly standard and easy to follow (I checked most of the theorems except for Lemma A.2 and the results for stochastic gradients). The authors do not clearly contrast their results to prior work (especially Bartlett et al. (2019) and Arora et al. (2019a)) and I’m therefore not convinced the final result brings any new insight. Please address this issue in your rebuttal. I will reconsider my score if the authors can provide a satisfactory answer.

Comparison to Du & Hu
The authors claim that their bound shows an improvement by a factor of O(kappa L) over the deep linear network (without residual connections) analyzed in Du & Hu.
1) I don’t think this is explicitly stated in the paper but are the initialization conditions and the assumed distance to the optimum the same in both papers?
2) These results are obviously worst-case bounds and the analysis used in both papers is different to some extent. Couldn’t you re-derived this result by adapting your own analysis?
3) Is there any theoretical proof or empirical evidence showing that resnets do indeed scale better w.r.t. to the condition number of the data?

Comparison to prior work
The authors mention the work of Bartlett et al. (2019) and Arora et al. (2019a) but it is never very clear what the real differences are.
1) Regarding Bartlett et al. (2019), you say that “Theorem 3.1 can imply the convergence result in Bartlett et al. (2019).”. Does the result of Theorem 3.1 provides a tighter bound in terms of L or kappa (when using the same initialization conditions)?
2) Arora et al. (2019a): You explain that they showed that “GD converges under substantially weaker conditions”. How much weaker are these conditions? How do they compare to the “modified identity transformation” that allows you to obtain a global convergence rate. How does your final result compare to Arora et al. (2019a)? The analysis of Arora et al. (2019a) requires a balanced-ness condition, is this substituted by a different condition in your analysis?
3) Finally, Allen-Zhu et al. (2019) already has some results on deep resnets although their analysis is for a heavily over-parametrized regime. Can you still comment on how their results differ from yours?

A & B are fixed matrices initialized from a Gaussian distribution. How essential is this condition? Assuming for simplicity that A and B are square matrices. Can I not set A=B=identity and still satisfy the condition in Theorem 1? The term on the RHS would be 1 so then the initialization would need to scale as a function of the spectrum of the data matrix X.

ResNet vs LinearNet
Where does your proof break down for a linear net without residual connections, i.e. where does the proof absolutely need the identify matrices. Looking at the proof, it seems to me, one could change the following in order to still obtain a similar result:
1) If we consider a network B \prod_l W_l AX, the proof of proposition 3.3 could be unchanged if the W matrices are initialized to identify instead of zero.
2) The lower bound on \sigma^2_min(I + \tau W_l) would instead be replaced by a lower on \sigma^2_min(W_l)=\lambda_min(W_l). Is this where  one can see the benefit of having residual connections?

Proposition 3.3
The bound on the singular values of the matrices A and B is vacuous for square matrices, in which case I believe the theorem does not hold. Can you comment on this?

Extension stochastic setting
I only skimmed at the proof but the extension looks fairly straightforward. Can you comment on how difficult this derivation is compared to the deterministic setting?

Extensions
You claim "can potentially provide meaningful insights to the convergence analysis of deep non-linear ResNets." although this is not obvious as the landscape of such networks can be very different. Can you elaborate?


**Experience Assessment:**

I have published one or two papers in this area.

**Review Assessment: Checking Correctness Of Derivations And Theory:**

I carefully checked the derivations and theory.

**Review Assessment: Checking Correctness Of Experiments:**

I assessed the sensibility of the experiments.

**Review Assessment: Thoroughness In Paper Reading:**

I read the paper thoroughly.

---

> ### Author Response · Authors · 2019-11-11
> **Response to Review #1: part 1**
>
> Thanks for your valuable and helpful comments.
>
> Q1:  “I don’t think this is explicitly stated in the paper but are the initialization conditions and the assumed distance to the optimum the same in both papers?”
> A1: Regarding the initialization, we consider random Gaussian transformations at both input and output layers and use zero initialization for all rest layers (which is equivalent to identity initialization for standard deep linear networks), while Du&Hu consider random Gaussian initialization for all layers. Therefore, the initialization scheme in our paper is different from that in Du&Hu.  A main finding of our paper is that the initialization considered here enables much stronger convergence guarantees.  In addition, both our paper and Du & Hu do not assume the distance to the optimum. We would also like to point out that the global convergence of GD (and SGD) established in our paper and in  Du&Hu do not require any condition on the distance to the optimum.
>
> Q2: “Couldn’t you re-derived this result by adapting your own analysis?”
> A2: We believe that the techniques of our paper could be generalized to other initialization schemes, such as the one considered by Du & Hu.
>
> Q3: “Is there any theoretical proof or empirical evidence showing that resnets do indeed scale better w.r.t. to the condition number of the data?”
> A3: We have proved a stronger upper bound on the required width in terms of the condition number;  we agree that lower bounds for other initialization schemes, or experimental comparisons, would be an interesting topic for future work.
>
>
> Q4: “Does the result of Theorem 3.1 provides a tighter bound in terms of L or kappa (when using the same initialization conditions)? ”
> A4:  Bartlett et al, only consider the case in which the hidden layers have the same width as the input and the output, so we cannot compare our bounds on the required width with theirs.  Our remark was that we can recover the main result of Bartlett, et al’s paper by specializing Theorem 3.1 to the case that $A=B=I$, and all hidden layers have the same number of hidden units as the input and output layers.
>
> Q5.1: “Arora et al. (2019a): You explain that they showed that “GD converges under substantially weaker conditions”. How much weaker are these conditions?”
> A5.1: We would like to first clarify that the statement “ Arora et al. (2019a) proves global convergence of GD under substantially weaker conditions”  is made for the comparison between  Arora et al. (2019a) and Bartlett et al. (2019), rather than our paper. Specifically, in order to establish the global convergence of GD, Arora et al. (2019a) requires (1) the widths of hidden layers are greater than or equal to the minimum between the dimensions of input and output; (2) layers are initialized to be approximately balanced; and (3) the initial loss is smaller than any loss obtainable with rank deficiencies. These conditions are weaker than that in Bartlett et al. (2019) as Bartlett et al. (2019) requires that the initial loss is much smaller. In contrast to Arora et al. (2019a) and Bartlett et al. (2019), our result does not require any assumption on the initial loss, and therefore holds under weaker conditions.
>
> Q5.2: “How do they (Arora et al. 2019a) compare to the “modified identity transformation” that allows you to obtain a global convergence rate.  How does your final result compare to Arora et al. (2019a)? ”
> A5.2: We would like to clarify that the global convergence of GD/SGD established in our paper is not limited to “modified identity transformation”. In fact, our paper proves a generic condition for input and output linear transforms such that the global convergence can be guaranteed. For example, in Corollaries 3.4, 3.7, 3.9, we show that Gaussian random linear transformation can guarantee global convergence. In addition, we also want to point out that the proposed modified identity transformation is just a possible choice of A and B (instead of random A and B) that can guarantee the global convergence of GD/SGD. In particular, as long as the width of the neural network is greater than or equal to $d+k$, modified identity transformation satisfies the aforementioned generic condition and therefore the global convergence of GD/SGD also holds. We have added this result in Section 4 of our revision.
> As we mentioned in A5.1, since our convergence guarantee does not require any condition on the initial loss while the result in Arora et al. (2019a) requires, we believe our result is stronger than that in Arora et al. (2019a).
>
> Q5.3: “The analysis of Arora et al. (2019a) requires a balanced-ness condition, is this substituted by a different condition in your analysis?”
> A5.3: Since in our paper all the hidden layers are simply initialized by identity matrix, they automatically satisfy the balanced-ness condition. We have clarified this in Section 2 of our revision.

---

> > ### Comment · AnonReviewer1 · 2019-11-13
> > **Q5 (Arora et al.)**
> >
> > You said "the initial loss is smaller than any loss obtainable with rank deficiencies."
> > Are you referring to the condition on T in Theorem 1 in https://arxiv.org/pdf/1810.02281.pdf? Please clarify.
> >
> > The theorem from Arora et al. also requires a condition on the margin deficiency of the initial weights, which is trivially met due to your initialization. You do have an additional condition on the input and output weights matrices, but this is not needed in Arora et al. if we consider the zero initialization of the weight matrices, can you please confirm?

---

> > > ### Author Response · Authors · 2019-11-15
> > > **Re:  Q5 (Arora et al.)**
> > >
> > > Thanks for your further comments.
> > >
> > > **Clarification on the initial loss condition:
> > >
> > > First, we would like to clarify that the statement “the initial loss is smaller than any loss obtainable with rank deficiencies” is quoted from the introduction of https://arxiv.org/pdf/1810.02281.pdf (see the third paragraph on page 2 for details). Second, this condition actually says that $\|W_{1:N}(0)-\Phi\|_F\le \sigma_{min}(\Phi)-c$, as stated in the first sentence of Theorem 1 in  Arora et al. (2018), which is not the condition on T in Theorem 1.
> > > Third, we do not see that our initialization trivially has a positive deficiency margin.  For example, for the case of the modified identity transformations for $A$ and $B$, we have $B(I + \tau W_{L}^{(0)})\dots (I + \tau W_{1}^{(0)})A = 0$. Then the margin deficiency condition would require $\| B(I + \tau W_{L}^{(0)})\dots (I + \tau W_{1}^{(0)})A -\Phi \|_F = \|\Phi \|_F \leq \sigma_{min}(\Phi)-c$ for some positive $c$, which apparently doesn't hold. So our initialization scheme does not necessarily satisfy margin deficiency condition imposed in Arora et al. (2018).
> > >
> > >
> > > **Comparison with Arora et al. in terms of the condition on the input and output weight matrices:
> > >
> > > It is true that we require certain conditions on the input and output weights matrices, which are stated in all theorems. However, Arora et al. 2018 also require conditions on the input and output weights matrices, i.e., balancedness condition (actually this condition is required for all layers in Arora et al. 2018, while in our paper, as we answered in A5.3, only the weight matrices at hidden layers satisfy this condition.) To our knowledge, the balancedness condition and our condition on the input and output weight matrices are not directly comparable, so it is unfair to say that we require “additional” condition to prove the global convergence.

---

> ### Author Response · Authors · 2019-11-11
> **Response to Review #1: part 2**
>
> Q6: “Finally, Allen-Zhu et al. (2019) already has some results on deep resnets although their analysis is for a heavily over-parametrized regime. Can you still comment on how their results differ from yours?”
> A6: It is true that Allen-Zhu et al. (2019) studied the convergence of ResNets for nonlinear activation functions. Besides the linear versus nonlinear activation functions, the additional differences include: (1) they require that the neural network to be vastly over-parameterized  and the width has a worse dependency on the sample size n and depth L, while our results for deep linear ResNets give a much better condition on the width (only logarithmic in n and with no dependence on L); and (2) Allen-Zhu et al. (2019) require that all data points are separated by a positive distance and have unit norm, our results have no assumption on the training data. We have added more comments in the related work of our revision.
>
>
> Q7.1: “A & B are fixed matrices initialized from a Gaussian distribution. How essential is this condition? Assuming for simplicity that A and B are square matrices. ”
> A7.1: First, we would like to clarify that this condition is not essential and Gaussian linear transformations are just one example we discussed. Second, while we have not worked out the details, we believe our theory can also be established if A and B are also trained. The high-level idea to prove this is: with proper initialization of A and B (e.g., Gaussian initialization with proper variance), we can show that they will stay close to their initial values similarly to the way that other layers were treated, and, given that they are close to their initial values, their singular values can be bounded above, and bounded away from zero, so that the objective function stays smooth, and they continue to propagate gradients effectively to the other layers.
>
> Q7.2: “Can I not set A=B=identity and still satisfy the condition in Theorem 1? The term on the RHS would be 1 so then the initialization would need to scale as a function of the spectrum of the data matrix X.”
> A7.2: Theorem 3.1 establishes convergence for any choices of A and B that satisfy a condition relating their singular values to the quality of the initial solution obtained from them. We have discussed the case that A=B=I in Remark 3.2 and Section 4. In this case, Bartlett, et al. showed that GD sometimes fails to converge to the optimum.
>
>
> Q8:  ”1) If we consider ...  initialized to identify instead of zero.
> 2) The lower bound ... . Is this where  one can see the benefit of having residual connections?“
> A8: Yes, you are right. We have already mentioned that our initialization is equivalent to the identity initialization for standard deep linear networks in Section 2. Thus it is also true that the  $\sigma^2_{min}(I + \tau W_l)$ can be replaced by $\lambda_{min}(W_l)$. We choose to present our results in the context of “deep linear ResNets” because we want to highlight the benefit of having residual connections.
>
> Q9: ”The bound ... the theorem does not hold. Can you comment on this?“
> A9: In Proposition 3.3, we require that $m\ge C(d+k+\log(1/\delta))$ for some absolute constant C, which means that A and B cannot be square matrices since $m\ge d$ and $m\ge k$. Then our theorem can indeed hold since the smallest singular values of A and B can be lower bounded with high probability.
>
> Q10: ”Can you comment on how difficult this derivation is compared to the deterministic setting?“
> A10: As we pointed out in the introduction section, we need to guarantee that the restricted gradient bounds and a smoothness property hold along the algorithm trajectory. Compared with the deterministic setting, the trajectory of SGD is difficult to characterize due to the randomness caused by stochastic gradients. Therefore, a martingale based proof technique is needed to characterize the trajectory of SGD in order to prove the global convergence. In addition, it is also highly non-trivial to prove the linear rate of convergence for SGD as we did in Theorem 3.8 because in general SGD cannot achieve this.
>
> Q11: ”You claim ... this is not obvious as the landscape of such networks can be very different. Can you elaborate?“
> A11: Deep linear networks are an idealized setting that includes some aspects of the non-linear case, including the potential for exploding and vanishing gradients, and a non-convex objective function, while abstracting away others.  Study of this clean setting has inspired research into the nonlinear case in the past.  The fact that stronger results are possible for residual networks in the linear case raises the hope that they may lead to stronger results with non-linearities.  We also were surprised by the results on effective initialization in this case discussed in Section 4;  we hope (and expect) that similar initialization schemes can be shown to work very efficiently in the presence of non-linearities.

---

> > ### Comment · AnonReviewer1 · 2019-11-13
> > **A11**
> >
> > Regarding your answer to Q11: Can you elaborate on this? I was expecting a slightly more technical answer. Take for instance Lemma A.2: looking at the  proof, it seems to me it does not require smoothness so it could potentially be adapted to non-linear networks? What would then be the difficulty in the initialization scheme that you mentioned in your answer?

---

> > > ### Author Response · Authors · 2019-11-15
> > > **Re: A11**
> > >
> > > Thank you for your further comments.
> > >
> > > We would like to clarify that Lemma A.2 implicitly requires smoothness since we directly leverage the formula of the loss function (which is smooth) in the proof. For non-linear networks there are generally two cases: using smooth activation functions (e.g., sigmoid, softplus, ELU) or nonsmooth activation functions (e.g., ReLU, LeakyReLU). For the smooth activation function case, we can easily derive a similar result as Lemma A.2. For the non-smooth activation function such as ReLU, we can prove a variant of Lemma A.2, which states the semi-smoothness property of the training loss function, as what was done in Allen-Zhu et al. (2019) and Frei et al. (2019).
> > >
> > > Regarding extending the analysis of our initialization scheme to nonlinear ResNets, the key is to prove two major results based on our initialization: 1) Residual connection with zero initialization can stabilize the forward feature propagation across layers, and 2)  Using Residual connection with zero initialization, one can provide a better characterization of the backward error propagation across layers, and a more refined optimization analysis in terms of all hidden layers, which cannot be easily done for deep nonlinear networks without residual connection.  We believe that by proving these two main results, the benefit of residual connection with zero initialization can be adapted to non-linear ResNets.

---

### Official Review · AnonReviewer2 · 2019-10-24
**Official Blind Review #2**

**Rating:** 6

**Review:**


*Summary*
This paper deals with the global convergence of deep linear ResNets. The author show that under some initialization conditions for the first and the last layer (that are not optimized !) GD and SGD does converge to a global minimum of the min squared error. The closed related work seems to be Bartlett et al. 2019 that study the convergence of GD in the case of linear networks.

*Decision*
On issue for Bartlett et al. 2019 was that they required a condition on the initial suboptimality to be small in order to insure convergence. This work shows that in the case of linear ResNets, with a well chosen initialization, a similar condition holds with high probability.
I think this paper is interesting for the ICLR community and seems to provide good contributions (like for instance the analysis for SGD). However I have some question that I would like the authors to answer.
*Questions*
- In Proposition 3.3 you show an upperbound on $\sigma_{\min}(B)$ in order to show in Corollary 3.4  that the condition to apply Theorem 3.1 is true. However, it seems to me that you need a lower bound on  $\sigma_{\min}(B)$ to prove that (A.3) is true.
- To what extent the proof of Theorem 3.1 uses the proof technique of Bartlett et al. 2019 ?
- With you small enough conditions, are you in the lazy regime described by Chizat, Lenaic, Edouard Oyallon, and Francis Bach. "On Lazy Training in Differentiable Programming." (2019). NeurIPS
- In Theorem 3.1 What is $e$ ?
- Could you prove the same result as Theorem  3.1 and 3.6 but with an inequality constraint on the step size? It seems very restrictive to me to ask a stepsize to be exactly equal to a quantity. If you cannot relax you equality constraint into an inequality constraint, can you at least show that your result hold for step size in an interval?


=== After rebuttal ===
Thank you for this detailed answer. It confirms that this paper is of interest to the ICLR community.


**Experience Assessment:**

I have read many papers in this area.

**Review Assessment: Checking Correctness Of Derivations And Theory:**

I assessed the sensibility of the derivations and theory.

**Review Assessment: Checking Correctness Of Experiments:**

I assessed the sensibility of the experiments.

**Review Assessment: Thoroughness In Paper Reading:**

I read the paper at least twice and used my best judgement in assessing the paper.

---

> ### Author Response · Authors · 2019-11-11
> **Response to Review #2**
>
> Thanks for your positive and constructive comments.
>
> Q1: “In Proposition 3.3 you show an upper bound on $\sigma_{min}(B)$ in order to show in Corollary 3.4  that the condition to apply Theorem 3.1 is true. However, it seems to me that you need a lower bound on  to prove that (A.3) is true.”
> A1: Thank for catching this.  We have actually proved the lower bound of $\sigma_{min}(B)$ in the proof of Proposition 3.3, but we did not properly present it in the statement of Proposition 3.3. We have modified the statement of Proposition 3.3 in the revision.
>
> Q2: “To what extent the proof of Theorem 3.1 uses the proof technique of Bartlett et al. 2019 ?”
> A2: There are a number of significant differences, but we would like to highlight Lemma A.2, which establishes the smoothness of the objective function.  Bartlett, et al established smoothness through an upper bound on operator norm of the Hessian in terms of parameters of the problem like the depth and the size of the input that held for all solutions inside a ball centered at the initial solution.  This was in turn used to bound how much the objective function changes as GD makes its step.  We instead directly bound this change in the objective.  Crucially, our bound on the smoothness parameter is tighter, which depends on the loss function value at a specific $\mathbf{W}$ (See the term $\sqrt{eL(\mathbf{W})}$ on the R.H.S. of the equation in Lemma A.2.).  This gives us a stronger bound after training has proceeded for a while ($L(\mathbf{W})$ is decreasing when the training makes progress).  We also achieved a quantitatively stronger dependence on the other parameters (e.g., input dimension $d$).
>
>
> Q3: “With you small enough conditions, are you in the lazy regime described by Chizat, Lenaic, Edouard Oyallon, and Francis Bach. "On Lazy Training in Differentiable Programming." (2019). NeurIPS”
> A3: Thanks for pointing this out. We would like to clarify that when applying Gaussian random transformations and the neural network width goes to infinity, our results are indeed in the lazy regime since the neural network weights will barely change. However, our theoretical results are not limited to this. Specifically, we provide a generic condition on the linear transformations at input and output layers, which can be satisfied in multiple ways.  For example, if the initial training loss is close to the global optimum, we do not require super large m, and the neural network does not need to behave like a linear model, which is not in the lazy regime. We have commented on this work in the additional related work section of our revision.
>
> Q4: “In Theorem 3.1 What is e?”
> A4: It is the base of the natural logarithm.
>
> Q5: “Could you prove the same result as Theorem  3.1 and 3.6 but with an inequality constraint on the step size? It seems very restrictive to me to ask a stepsize to be exactly equal to a quantity. If you cannot relax you equality constraint into an inequality constraint, can you at least show that your result holds for step size in an interval?”
> A5: Thanks for your suggestion. Actually, the quantities we provide in Theorem 3.1 and 3.6 are only the upper bound of step size (which corresponds to the optimal iteration complexity), our theoretical results still hold for the smaller step size. We have restated the condition on the stepsize to be $\eta \leq \dots$ in all theorems of the revision.

---

> > ### Comment · AnonReviewer2 · 2019-11-14
> > **Thank you for your response**
> >
> > Thank you for this detailed answer. It confirms that this paper is of interest to the ICLR community.

---

### Decision · Program_Chairs · 2019-12-19

**Decision:**

Accept (Poster)

**Comment:**

This paper provides further analysis of convergence in deep linear networks. I recommend acceptance.